# Fantastic Rewards and How to Tame Them: A Case Study on Reward Learning for Task-oriented Dialogue Systems

**Yihao Feng**[*1], **Shentao Yang**[*2], **Shujian Zhang**[2], **Jianguo Zhang**[1]
**Caiming Xiong**[1], **Mingyuan Zhou**[2], **Huan Wang**[1]
[1] Salesforce Research  [2] The University of Texas at Austin

## Abstract

When learning task-oriented dialogue (ToD) agents, reinforcement learning (RL) techniques can naturally be utilized to train dialogue strategies to achieve user-specific goals. Prior works mainly focus on adopting advanced RL techniques to train the ToD agents, while the design of the reward function is not well studied. This paper aims at answering the question of *how to efficiently learn and leverage a reward function for training end-to-end (E2E) ToD agents*. Specifically, we introduce two generalized objectives for reward-function learning, inspired by the classical learning-to-rank literature. Further, we utilize the learned reward function to guide the training of the E2E ToD agent. With the proposed techniques, we achieve competitive results on the E2E response-generation task on the Multiwoz 2.0 dataset. Source code and checkpoints are publicly released at https://github.com/Shentao-YANG/Fantastic_Reward_ICLR2023.

## 1 Introduction

The bloom of pre-training language models (*e.g.*, Devlin et al., 2018; Lewis et al., 2019; Radford et al., 2019; Zhang et al., 2022c) have significantly pushed the boundaries of natural language processing (NLP) on real-world tasks. Among all the promising potentials, one important application is the task-oriented dialogue (ToD) systems, which interact with the users in multiple turns via natural languages to accomplish tasks such as weather inquiry, ticket booking, or schedule planning (Chen et al., 2017; Kwan et al., 2022).

Traditionally, the problem of ToD is decomposed into several sub-tasks (Smith & Hipp, 1994; Young et al., 2013): natural language understanding (NLU) for understanding turn-level user intents or slot values (Tur & De Mori, 2011; Casanueva et al., 2020), dialogue state tracking (DST) for tracking user belief state across multiple dialogue turns (Zhang et al., 2019; Zhu et al., 2020), dialogue management (DM) for choosing system actions to take (Peng et al., 2017; Zhao et al., 2019), and natural language generation (NLG) for mapping system actions to natural language responses (Wen et al., 2015; Zhang et al., 2020). This pipeline approach, however, requires intensive structural designs and comprehensive data annotation for model training (Kwan et al., 2022). Recently, there has been a growing interest in building end-to-end (E2E) ToD agents, which directly generate responses based on the natural language conversation mixing user utterances and past responses. Apart from this structural simplicity, many of the E2E ToD models can utilize the pre-trained language models and are simply trained by supervisedly fine-tuning the pre-trained models on the ToD datasets (*e.g.*, Hosseini-Asl et al., 2020; Ham et al., 2020; Lin et al., 2020; Peng et al., 2021).

Due to the intrinsic similarity between dialogues and sequential decision-making, reinforcement learning (RL) methods are naturally employed to train dialogue systems and have achieved some success (*e.g.*, Williams & Young, 2007; Georgila & Traum, 2011; Zhao et al., 2019). Since interacting with users during the training process is mostly impractical, offline RL (Lange et al., 2012; Levine et al., 2020), *i.e.*, RL on static datasets, has recently been adopted to train E2E ToD models (*e.g.*, Jaques et al., 2019; 2020; Ramachandran et al., 2021; Snell et al., 2022a;b; Jang et al., 2022). Although this direction already presents promising empirical results, an open question exists on how to properly design the reward function for the underlying (offline) RL. Existing works (*e.g.*,

---

*Equal Contribution. Corresponds to {yihao.ac@gmail.com, shentao.yang@mccombs.utexas.edu}.

Wu et al., 2019c; Jang et al., 2022; Snell et al., 2022b) manually design a sparse reward function that only indicates whether the agent achieves the goal or not. Unfortunately, due to the delayed feedback, learning from such a sparse reward signal is itself challenging for RL agents (Andrychowicz et al., 2017; Liu et al., 2019; Durugkar et al., 2021). When applied to train the more complicated ToD agents, the sparse reward signal could lead to poor empirical performance (Takanobu et al., 2019; Wang et al., 2020a). To address this issue, we aim at answering the following question in this paper:

> How to efficiently **learn** a reward function and **leverage** it for training E2E dialogue agents?

We answer the *first half of this question* by introducing two reward-learning objectives, RewardNet and RewardMLE, based on the classical learning-to-rank literature (Cao et al., 2007; Xia et al., 2008). Our desiderata is a reward function that can "explain" some non-trivial preference-based ordering among multiple alternative dialogue trajectories, thus potentially allowing the resulting RL-trained ToD agents to have better-than-demo performance. We accomplish this goal by learning a parameterized reward function on dialogue turns, from which the accumulated reward of a dialogue trajectory can reflect the preference among multiple alternatives. We answer the *second half of the question* by utilizing the learned reward function to guide the training of the E2E ToD system, with special considerations on the training stability. With these answers to the above question, we achieve competitive results on the E2E response-generation task on the widely-used dialogue benchmark MultiWOZ 2.0 (Budzianowski et al., 2018). Several ablation studies and analyses are conducted to provide further insights into the proposed techniques.

## 2 BACKGROUND

**Task-oriented dialogue as reinforcement learning.** We formulate the problem of the ToD system as a partially observable Markov decision process (POMDP) (Kaelbling et al., 1998), specified by $\mathcal{M} = \langle \mathbb{S}, \mathbb{A}, \mathbb{O}, \mathcal{P}, \mathcal{R}, \gamma \rangle$, where state $s \in \mathbb{S}$ consists of the previous dialogue history $h$ and the user intended goal $g$ specified prior to the start of the dialogue; $o \in \mathbb{O}$ is the observation that can be the user utterance; action $a \in \mathbb{A}$ can be the system response or dialogue act; $\mathcal{P}(s' \,|\, s, a)$ is the underlying transition probability; $\mathcal{R}(h, a, g)$ is the intermediate reward function for taking action $a$ under dialogue history $h$ and goal $g$; and $\gamma \in [0, 1]$ is the discount factor.

The dialogue history $h_t$ at timestep $t$ consists of all the previous observations and actions, *i.e.*, $h_t \triangleq \{o_0, a_0, \dots, o_{t-1}, a_{t-1}, o_t\}$. Since the ToD agent cannot directly observe the user goal $g$, it makes a decision based on the entire dialogue history $h_t$ so far. Specifically, the policy $\pi$ is defined as a mapping from $h_t$ to a probability distribution over $\mathbb{A}$, *i.e.*, $\pi \triangleq \pi(a_t \,|\, h_t)$. The training objective is to find a policy $\pi$ that maximizes the expected (discounted) cumulative reward

$$J(\pi) \triangleq \mathbb{E}_{\mu_g, \pi, \mathcal{P}} \left[ \sum_{t=0}^{T} \gamma^t \mathcal{R}(h_t, a_t, g) \right] ,$$

where $\mu_g$ is the distribution of goals and $T$ is the number of turns in the dialogue trajectory.

**Reward design and learning in ToD systems.** Unlike the classical RL problems where the intermediate reward function is well designed and provided, in ToD systems we can only get the evaluation results at the end of the dialogue (Budzianowski et al., 2018). Consequently, most of the existing works adopt the manually designed intermediate reward function that only gives binary reward to indicate whether the dialogue agent achieves the goal or not (*e.g.*, Weisz et al., 2018; Wu et al., 2019c; Jang et al., 2022):

$$\mathcal{R}(h_t, a_t, g) = \begin{cases} R_{\text{const}} \text{ or } 0, & \text{if goal } g \text{ is achieved at timestep } t \,, \\ -R_{\text{const}}, & \text{if goal } g \text{ is not achieved at timestep } t \,, \end{cases}$$

where $R_{\text{const}}$ is a positive constant that can be 1. However, such a sparse reward signal can be one of the reasons that the ToD agents from RL often have poor performance (Takanobu et al., 2019; Wang et al., 2020a). A similar issue is also observed in goal-oriented RL (Andrychowicz et al., 2017).

To address the above issue, a few recent works focus on learning an intermediate reward function from demonstrations or mechanical dialogue assessments (*e.g.*, Wang et al., 2020a; Ramachandran et al., 2021), inspired by the reward-learning-from-preferences in RL (*e.g.*, Christiano et al., 2017; Brown et al., 2019; 2020). More precisely, suppose we are given two dialogue trajectories $\tau_i$ and $\tau_j$, taking the form $\tau_i \triangleq \{g^{(i)}, (o_0^{(i)}, a_0^{(i)}), \dots, (o_T^{(i)}, a_T^{(i)})\}$. We want to learn a

parametrized reward function $\mathcal{R}_\theta(o_t, a_t, g)$ with parameter $\theta,$[1] such that $\sum_{t=0}^T \mathcal{R}_\theta(o_t^{(i)}, a_t^{(i)}, g^{(i)}) > \sum_{t=0}^T \mathcal{R}_\theta(o_t^{(j)}, a_t^{(j)}, g^{(j)})$ when $\tau_i$ is preferred over $\tau_j$ (denoted as $\tau_i \succ \tau_j$) and *vice versa*. Then one can follow the Bradley-Terry model of pairwise preferences (Bradley & Terry, 1952) to train the reward function by minimizing the loss

$$\ell(\theta) = -\sum_{\tau_i \succ \tau_j} \log \left[ \frac{\exp\left(\sum_{t=0}^T \mathcal{R}_\theta(o_t^{(i)}, a_t^{(i)}, g^{(i)})\right)}{\sum_{k \in \{i,j\}} \exp\left(\sum_{t=0}^T \mathcal{R}_\theta(o_t^{(k)}, a_t^{(k)}, g^{(k)})\right)} \right] . \tag{1}$$

$\ell(\theta)$ can be interpreted as a pairwise ranking loss, which is formalized as binary classification in the problem of learning to rank (Herbrich et al., 1999; Freund et al., 2003; Burges et al., 2005).

## 3  MAIN METHOD

In this section, we first introduce two objectives for reward-function learning based on the classical approaches in the learning-to-rank (LTR) literature (Liu, 2009). Then we use MinTL (Lin et al., 2020) as an example to demonstrate how we can use the learned reward function as a plugin module to improve existing methods of training the E2E ToD models.

### 3.1  TWO GENERALIZED OBJECTIVES FOR REWARD LEARNING

We introduce two objectives, `RewardNet` and `RewardMLE`, both of which can utilize multiple dialogue trajectories on each update for optimizing the reward function. Our motivation is that, compared with the pairwise approach described in Eq. (1), these two objectives consider more information at each training step, and thus can be more effective for reward learning and may lead to a better solution under the stochastic training setting.

**Setup.**  Assume that there are $N \geq 2$ dialogue trajectories, denoted by $\mathcal{D}_N \triangleq (\tau_1, \tau_2, \ldots, \tau_N)$, and each trajectory $\tau_i$ has an automatic evaluation score $S(\tau_i)$.[2] For simplicity, we assume that these $N$ dialogue trajectories are of equal length $T$ and are already sorted by the automatic evaluation scores, *i.e.*, $\tau_1 \succ \tau_2 \succ \cdots \succ \tau_N$, or equivalently, $S(\tau_1) > S(\tau_2) > \cdots > S(\tau_N)$. We denote the accumulated reward of dialogue trajectory $\tau_i$ from $\mathcal{R}_\theta$ as $J(\tau_i; \theta) = \sum_{t=0}^T \mathcal{R}_\theta(o_t^{(i)}, a_t^{(i)}, g^{(i)})$. Our goal is to learn a reward function $\mathcal{R}_\theta(o, a, g)$ such that the accumulated rewards of those trajectories can reflect the ranking order, *i.e.*, $J(\tau_1; \theta) > \cdots > J(\tau_N; \theta)$.

**RewardNet.**  The proposed `RewardNet` objective for reward function learning is adapted from the *ListNet* loss (Cao et al., 2007) in the LTR literature. Specifically, given $N$ trajectories and their associated scores, we define the `RewardNet` loss as the cross entropy between $\{J(\tau_i; \theta)\}_{i=1}^N$ and $\{S(\tau_i)\}_{i=1}^N$:

$$\ell_{\texttt{RewardNet}}(\theta; \mathcal{D}_N) \triangleq -\sum_{i=1}^N P_S(\tau_i) \cdot \log\left(P_{J(\tau;\theta)}(\tau_i)\right) , \tag{2}$$

with

$$P_S(\tau_i) = S(\tau_i) / \left(\sum_{k=1}^N S(\tau_k)\right), \quad P_{J(\tau;\theta)}(\tau_i) = \Phi(J(\tau_i; \theta)) / \left(\sum_{k=1}^N \Phi(J(\tau_k; \theta))\right),$$

where $\Phi(\cdot)$ is a monotonic positive function defined on $\mathbb{R}^+$, and $P_S(\tau_i)$ is a normalized probability vector defined by the true evaluation scores of those $N$ trajectories. Note that when $N = 2$ and $\Phi$ is the identity function, `RewardNet` can be viewed as a *soft* version of the pairwise preference loss defined in Eq. (1), where the hard binary preference labels are replaced by $\{P_S(\tau_i)\}_{i=1}^N$. This soft pairwise loss is adopted for reward learning in the recent CASPI paper (Ramachandran et al., 2021).

**RewardMLE.**  The `RewardMLE` objective is based on the *ListMLE* loss (Xia et al., 2008), where we only utilize the ranking order in the batched dialogue trajectories $\mathcal{D}_N$, rather than the original metric scores $\{S(\tau_i)\}_{i=1}^N$. Let $y = \text{rank}(S)$ be the random variable that represents the ranking order of the dialogue trajectories ($y(\tau_i) = i, \forall i$, if the batched trajectories $\mathcal{D}_N$ are sorted). The `RewardMLE` objective is defined as the negative log-likelihood of the ranking order $y$ under the Plackett-Luce choice model (Plackett, 1975; Luce, 2012) induced by the accumulated reward of each trajectory $\{J(\tau_i; \theta)\}_{i=1}^N$. Specifically, the loss is defined as

$$\ell_{\texttt{RewardMLE}}(\theta; \mathcal{D}_N) \triangleq -\log P\left(y \mid \{J(\tau_i; \theta)\}_{i=1}^N\right) , \tag{3}$$

---

[1]We use the belief state, action, and goal as the reward function inputs. The belief state is part of the observation $o_t$. We also drop the dependency on $h_t$ for $\mathcal{R}_\theta$ to simplify the reward function learning.

[2]We use the `Combined Score` (Mehri et al., 2019) as $S(\tau_i)$. Detailed definition is delayed to Section 5.

with

$$P\left(y \,|\, \{J(\tau_i; \theta)\}_{i=1}^N\right) = \prod_{i=1}^N \left\{ \Phi(J(\tau_i; \theta)) \Big/ \sum_{k=i}^N \Phi(J(\tau_k; \theta)) \right\},$$

where trajectories in $\mathcal{D}_N$ are assumed sorted as described in the problem setup, *i.e.*, $\tau_1 \succ \cdots \succ \tau_N$. Since `RewardMLE` only requires the ranking information derived from the raw scores, it is potentially a more robust choice when the preference scores could be inaccurate.

In Eqs. (2) and (3), the monotonic positive function $\Phi$ transforms the unnormalized inputs $\{J(\tau_i; \theta)\}_{i=1}^N$ to a $N$-dimensional probabilistic simplex. In this work, we consider $\Phi$ being exponential function $\exp(\cdot)$ and power function $(\cdot)^p, p \in \mathbb{N}$, which are respectively known as the softmax transform and the escort transform (Mei et al., 2020).

## 3.2 Policy Gradient Update with Learned Reward Function

With the learned reward function $\mathcal{R}_\theta(o, a, g)$, the next step is to improve the parametrized dialogue agents $\pi_\phi$ via policy gradient methods (Sutton & Barto, 2018), given a collected offline dataset $\hat{\mathsf{D}} := \{\tau_k\}_{k=1}^K$. A classical approach to train the policy $\pi_\phi$ is to estimate the policy gradient via the REINFORCE method (Williams, 1992):

$$\nabla_\phi J_{\text{REINFORCE}}(\pi_\phi) = \mathbb{E}_{(g, h_t) \sim \hat{\mathsf{D}}, \tilde{a}_t \sim \pi_\phi(\cdot \,|\, h_t)}[\nabla_\phi \log \pi_\phi(\tilde{a}_t \,|\, h_t) \cdot G^{\pi_\phi}(h_t, \tilde{a}_t, g)], \qquad (4)$$

where $G^{\pi_\phi}(h_t, \tilde{a}_t, g)$ is the (discounted) accumulated reward that the agent $\pi_\phi$ receives, starting from the observation $o_t$ (part of $h_t$) and action $\tilde{a}_t$, under the given goal $g$. When the discount factor $\gamma > 0$, estimating $G^{\pi_\phi}(h_t, \tilde{a}_t, g)$ requires Monte Carlo sampling (on-policy) or temporal difference learning (off-policy), both of which require learning an additional value-function network. Empirically we observe that learning an additional action-value function could introduce instability and extra compute to the subsequent training of the E2E dialogue model. To simplify the training pipeline, we simply set the discount factor $\gamma = 0$, and thus $G^{\pi_\phi}(h_t, \tilde{a}_t, g) = \mathcal{R}_\theta(o_t, \tilde{a}_t, g)$.

Though the policy gradient estimator defined in Eq. (4) is unbiased, it tends to have high variance, especially when the action space is large. Unfortunately, in the E2E ToD system, the action space is often defined to be the Cartesian product of the vocabulary, which itself has a dimension larger than 30000. As a result, optimizing the agent $\pi_\phi$ by the REINFORCE estimator may suffer from divergent training. We illustrate this phenomenon via a toy example in Section 5.2.

To mitigate the high-variance issue of the REINFORCE estimator, we utilize the Gumbel-softmax (GS) trick (Jang et al., 2016; Maddison et al., 2016; Fan et al., 2021) to reduce the variance. Specifically,

$$J_{\text{GS}}(\pi_\phi) = \mathbb{E}_{a_t \sim \pi_\phi(\cdot \,|\, h_t)}[\mathcal{R}_\theta(o_t, a_t, g)] \approx \mathbb{E}_{\boldsymbol{\epsilon} \sim \text{Gumbel}(0,1)}[\mathcal{R}_\theta(o_t, f_\phi(h_t, \boldsymbol{\epsilon}), g)],$$

$$\text{with} \quad f_\phi(h_t, \boldsymbol{\epsilon}) = \left[ f_\phi^{(1)}(h_t, \boldsymbol{\epsilon}), \ldots, f_\phi^{(|\mathbb{A}|)}(h_t, \boldsymbol{\epsilon}) \right] \in \mathbb{R}^{|\mathbb{A}|}, \quad \text{and} \quad f_\phi^{(i)}(h_t, \boldsymbol{\epsilon}) = \frac{\exp((l_i(h_t; \phi) + \epsilon_i)/\lambda)}{\sum_{j=1}^{|\mathbb{A}|} \exp((l_j(h_t; \phi) + \epsilon_j)/\lambda)},$$

where $\{l_i(h_t; \phi)\}_{i=1}^{|\mathbb{A}|}$ are the logits of the categorical distribution defined by the agent $\pi_\phi$, and $\lambda$ is the temperature parameter that we set as 1. Besides, following the pessimistic principle in offline RL (Buckman et al., 2020), we add a weighted regularization such that the actions generated by the agent $\pi_\phi$ are close to the actions in the dataset $\hat{\mathsf{D}}$,

$$\ell_{\text{W}}(\pi_\phi) := -\mathbb{E}_{(h_t, a_t, g) \sim \hat{\mathsf{D}}}[\log \pi_\phi(a_t \,|\, h_t) \cdot \mathcal{R}_\theta(o_t, a_t, g)],$$

which is similar to the weighted behavior cloning in offline RL (Wang et al., 2020b), except that we directly use the intermediate rewards as the weights, rather than using the value function. Combining the policy gradient and the weighted regularization, we have the following loss for the agent $\pi_\phi$:

$$\ell_{\text{GEN}}(\phi) = -\alpha \cdot J_{\text{GS}}(\pi_\phi) + \ell_{\text{W}}(\pi_\phi), \qquad (5)$$

where $\alpha$ is the coefficient balancing these two parts. Note that the original supervised-learning loss of MinTL (Lin et al., 2020) can be decomposed into two parts, respectively for the dialogue state tracking (DST) and the response generation. We retain the DST loss $\ell_{\text{DST}}(\phi)$ in MinTL and replace its response-generation loss with Eq. (5). Our final loss for ToD agent training is

$$\ell(\phi) = \ell_{\text{GEN}}(\phi) + \ell_{\text{DST}}(\phi). \qquad (6)$$

We illustrate our method in Fig. 1 and provide an algorithm box in Appendix B.

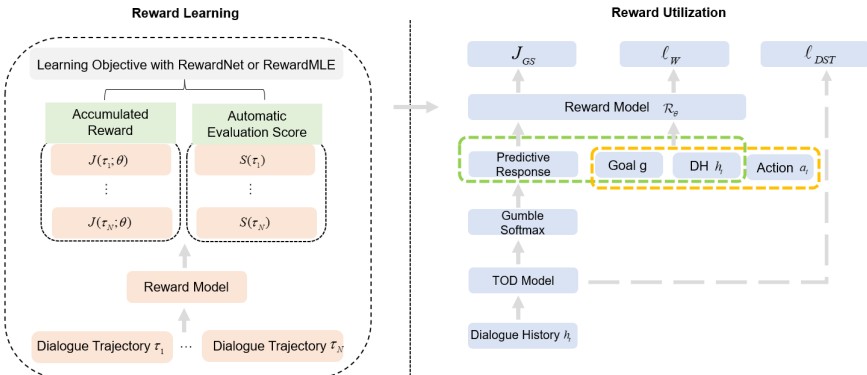

Figure 1: Overview of the proposed method. We denote "Accumulated Reward" for the learned accumulated reward, $J(\cdot; \theta)$ for the accumulated reward of each trajectory, $S(\cdot)$ for the combined score of each trajectory, $\ell_{\mathrm{W}}$ for the weighted regularization, $\ell_{\mathrm{DST}}$ for the DST loss, and "DH" for the dialogue history. In the right panel, $(h_t, a_t, g) \sim \hat{\mathsf{D}}$. We use BART for both the reward model and the ToD model.

**Remark** Eq. (6) for the learning of the dialogue agent $\pi_\phi$ is essentially a generalized objective from several previous works. Specifically, if we set $\alpha = 0$ and set the reward function to be constant $\mathcal{R}_\theta(o_t, a_t, g) \equiv 1$, Eq. (6) reduces to the objective in MinTL, without any guidance for response-generation from the learned reward function $\mathcal{R}_\theta$. If we set $\alpha = 0$, and use the RewardNet loss with $N = 2$ and $\Phi = (\cdot)^1$ (*i.e.*, the identity function) to train the reward function, Eq. (6) reduces to the objective in CASPI (Ramachandran et al., 2021). In Section 5, we demonstrate the advantages of our techniques proposed in this section, including the RewardNet and RewardMLE losses for reward learning, and the $J_{\mathrm{GS}}(\pi_\phi)$ for agent training.

## 4 RELATED WORK

Recent works on the E2E ToD systems (*e.g.*, Wu et al., 2019b; Lin et al., 2020; Hosseini-Asl et al., 2020; Ham et al., 2020; Peng et al., 2021; Yang et al., 2021) have significantly improved the overall system's performance and simplified the algorithmic designs in earlier works, which require solving several pipeline based sub-tasks (*e.g.*, Young et al., 2013; Gao et al., 2018; Zhang et al., 2020). The reward function trained by our methods can be leveraged as guidance to train existing E2E models, without changing the underlying structures. We demonstrate the effectiveness of our proposed reward learning methods under the structure of MinTL (Lin et al., 2020) and GALAXY (He et al. (2022); in Appendix E) where we only add an additional reward-function-guided objective for the response-generation model, while keeping other components of the respective structure unchanged.

One line of related research is applying RL to train ToD agents. It is often unsuccessful to directly apply RL algorithms such as the DDPG (Lillicrap et al., 2015) or PPO (Schulman et al., 2017) since the agent training could potentially diverge (Zhao et al., 2019; Jang et al., 2022; Kwan et al., 2022). Recently, a number of works consider offline RL (Levine et al., 2020) as a promising solution to stabilize the agent training on a static dataset (*e.g.*, Jaques et al., 2020; Ramachandran et al., 2021; Jang et al., 2022; Verma et al., 2022; Snell et al., 2022a;b). Following the offline RL principle, we use a reward-weighted regularization to stabilize the dialogue-agent training. Together with the incorporation of the Gumbel-softmax trick to estimate the policy gradient, our work retains algorithmic simplicity while improving the training stability and overall performance.

Finally, our paper closely relates to works on reward learning for the ToD systems (*e.g.*, Takanobu et al., 2019; Ramachandran et al., 2021). This research thread differs from works that directly use a manually designed reward function, which only gives sparse signals to indicate whether the agent achieves the goal or not (*e.g.*, Weisz et al., 2018; Wu et al., 2019c; Jang et al., 2022; Snell et al., 2022b). One line of this research direction is utilizing inverse reinforcement learning (IRL) (Russell, 1998) to learn a dense reward function, by assuming the collected data be expert demonstrations (Takanobu et al., 2019). However, modern IRL techniques such as GAIL-style algorithms (Ho & Ermon, 2016; Fu et al., 2017) often require iterating between reward learning and policy training (Finn et al., 2016), which is computationally expensive and less scalable to dialogue-generation models. Besides, the IRL methods aim at justifying the data, while the reward-learning framework in our work seeks to explain the preference among multiple trajectories, potentially leading to *better-than-demo* agents (Brown et al., 2019; 2020). Our paper is more closely related to the research on

Table 1: Results of the E2E response generation task on the MultiWOZ 2.0 dataset. The best result on each metric is bold. The results of UBAR are from the reproduction by Jang et al. (2022). The results of CASPI are from our reproduction. All our provided results are the average over five random seeds. Other results are from the original paper. "GS" denotes the Gumbel-softmax trick. $(\cdot)^1$ denotes the power function with power 1.

| Algorithms | Inform | Success | BLEU | Combined Score |
|---|---|---|---|---|
| SFN + RL (Mehri et al., 2019) | 73.80 | 53.60 | 16.90 | 83.10 |
| DAMD (Zhang et al., 2020) | 76.40 | 64.35 | 17.96 | 88.34 |
| SimpleTOD (Hosseini-Asl et al., 2020) | 84.40 | 70.10 | 15.01 | 92.26 |
| MinTL (Lin et al., 2020) | 84.88 | 74.91 | 17.89 | 97.78 |
| SOLOIST (Peng et al., 2021) | 85.50 | 72.90 | 16.54 | 95.74 |
| UBAR (Yang et al., 2021) | 87.47 | 74.43 | 17.61 | 98.56 |
| GPT-Critic (Jang et al., 2022) | 90.07 | 76.63 | 17.83 | 101.13 |
| CASPI[3] (Ramachandran et al., 2021) | 91.37 | 82.80 | 17.70 | 104.78 |
| RewardNet, $N = 3$, $\Phi = (\cdot)^1$ | 92.77 | 84.28 | 17.74 | 106.27 |
| RewardMLE, $N = 5$, $\Phi = \exp(\cdot)$ | 91.49 | 83.38 | **18.97** | 106.40 |
| RewardNet + GS, $N = 3$, $\Phi = (\cdot)^1$ | 92.63 | **84.32** | 18.35 | **106.83** |
| RewardMLE + GS, $N = 5$, $\Phi = \exp(\cdot)$ | **93.09** | 83.90 | 18.04 | 106.54 |

reward learning from preferences, which has recently been adopted in NLP tasks, including training language models (Fan et al., 2020; Ouyang et al., 2022) and fine-tuning (Ziegler et al., 2019; Zhang et al., 2021b; 2022b), question-answering with verification (Nakano et al., 2021; Zhang et al., 2021a; Menick et al., 2022; Zhang et al., 2022a), and ToD systems (Ramachandran et al., 2021). These works use the pairwise preference-learning objective in Christiano et al. (2017), which can be viewed as a special case of the RewardNet loss discussed in Section 3.1. Our work mainly focuses on the ToD task, where we study reward-function learning and reward utilization for training E2E dialogue agents. Appendix C continues the discussion on related reward-learning methods.

## 5 EXPERIMENTS

**Dataset.** We evaluate the proposed methods on the MultiWOZ 2.0 dataset (Budzianowski et al., 2018), which is a representative ToD benchmark. MultiWOZ 2.0 is a large-scale multi-domain dialogue corpus with seven domains: attraction, hospital, police, hotel, restaurant, taxi, and train. Each dialogue therein covers between one to three domains. This dataset has 8438 dialogues in the training set and 1000 dialogues in the validation and test set respectively.

**Evaluation Metrics.** Our proposed method is evaluated on the E2E dialogue-modeling task of the MultiWOZ 2.0 dataset. Following the standard setup (*e.g.*, Budzianowski et al., 2018; Mehri et al., 2019), we use four automatic evaluations metrics: 1) **Inform** rate: the fraction of the dialogues where the system has provided an appropriate entity; 2) **Success** rate: the fraction of the dialogues where the system answered all the requested information; 3) **BLEU** score (Papineni et al., 2002): measures the fluency of the generated responses; 4) **Combined Score** (Mehri et al., 2019): an overall quality measure defined as Combined Score =: (Inform + Success) × 0.5 + BLEU. Details on prepossessing and implementation are in Appendix B and G.

### 5.1 MAIN RESULTS

**Main evaluation.** Table 1 compares the performance of our methods with several classical and recent approaches in the E2E response-generation task. As shown in Table 1, our proposed methods not only improve the dialogue-task completion, measured by the Inform rate and the Success rate; but also generate fluent responses, reflected by the competitive BLEU scores. Recall that CASPI is a special case of the RewardNet loss (Eq. (2)) when we use escort transform ($\Phi = (\cdot)^1$, the identity function) with pairwise preference ($N = 2$). When we use three dialogue trajectories ($N = 3$) to construct the RewardNet loss and retain the same escort transform, the overall performance generally improves over CASPI. As discussed in Section 3.1, our RewardNet loss generalizes the pairwise-preference learning by taking more information on each update of the reward model and thus could learn a better reward function. Appendix D further compares our methods with CASPI.

The performance is further gently improved by changing the RewardNet loss (Eq. (2)) to the RewardMLE loss (Eq. (3)), with the softmax transform ($\Phi = \exp(\cdot)$) and $N = 5$ dialogue trajectories. This again demonstrates the benefit of our proposal of using multiple trajectories to learn the reward model. Section 5.2 conducts ablation studies on the number of trajectories and choice of $\Phi$.

---

[3]The CASPI paper reports the median score over random seeds, instead of the more commonly used mean score. We run the official CASPI codebase (https://github.com/salesforce/CASPI) and report the mean scores.

Table 2: Results on the simulated low-resource settings, where 5%, 10%, and 20% of the training data is used to train the models. The best result on each metric under each setting is bold. "Comb." is the Combined Score. All our provided results are the average over five random seeds. Baseline results are from Lin et al. (2020).

| Model | 5% | | | | 10% | | | | 20% | | | |
|---|---|---|---|---|---|---|---|---|---|---|---|---|
| | Inform | Success | BLEU | Comb. | Inform | Success | BLEU | Comb. | Inform | Success | BLEU | Comb. |
| DAMD | 56.60 | 24.50 | 10.60 | 51.15 | 62.00 | 39.40 | 14.50 | 65.20 | 68.30 | 42.90 | 11.80 | 67.40 |
| MinTL | 75.48 | 60.96 | 13.98 | 82.20 | 78.08 | 66.87 | **15.46** | 87.94 | 82.48 | 68.57 | 13.00 | 88.53 |
| RewardNet :3 | 81.22 | 67.37 | 12.82 | 87.11 | **92.39** | **78.98** | 13.36 | **99.05** | 89.83 | **79.30** | 15.18 | 99.75 |
| RewardMLE :5 | **82.90** | **69.61** | **14.26** | **90.51** | 89.67 | 77.48 | 14.80 | 98.38 | **90.15** | 78.70 | **15.81** | **100.24** |

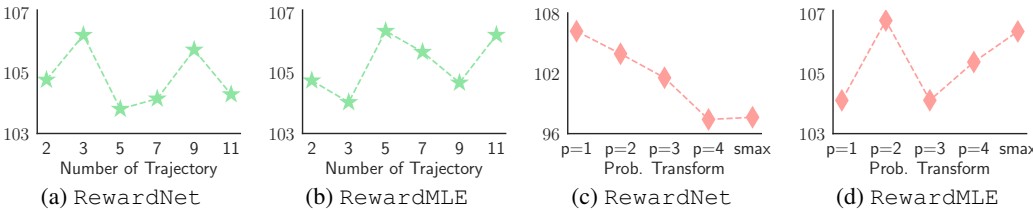

(a) RewardNet     (b) RewardMLE     (c) RewardNet     (d) RewardMLE

Figure 2: Line plots comparing the Combined Score when the RewardNet and RewardMLE losses are constructed under a different number of sampled trajectories or different probabilistic transforms. The $y$-axis represents the Combined Score. $p = 1, 2, 3, 4$ is the escort transform with power $1, 2, 3, 4$. "smax" is the softmax transform. Results are the average over five random seeds.

So far, we follow the prior work to not utilize policy gradient to train the response-generation model, *i.e.*, $\alpha = 0$ in Eq. (5). Extra performance gain can be obtained by adding the policy-gradient updates via the Gumbel-softmax trick (GS) discussed in Section 3.2. Indeed, GS improves both the plain RewardNet and RewardMLE models. This shows the efficacy of directly optimizing the response-generation model *w.r.t.* the learned reward function. Further discussion is provided in Section 5.2.

Appendix E provides the experimental results when applying our method onto the recent GALAXY backbone (He et al., 2022). Appendix F discusses the results on the MultiWOZ 2.1 dataset.

**Low-resource experiment.** We evaluate our method on the low-data regime by following the testing strategy in Lin et al. (2020). Specifically, we use 5%, 10%, and 20% of the training data to train our basic RewardNet and RewardMLE models in Table 1, without the GS component. We compare them with the baseline scores in Lin et al. (2020). Table 2 reports the results. It is clear that our models outperform the baselines, MinTL and DAMD, showing the efficacy of our method. Compared with Table 1, our models trained with 20% of the data perform competitively with the baseline methods trained on the full training set.

## 5.2 ABLATION STUDY

The ablation study considers the following four research questions to better understand our methods.

**(a):** *What if we learn the reward function via a different number of trajectories?* In Fig. 2a and 2b, we vary the number of trajectories used for the reward-learning losses in Table 1. To avoid unwanted interference, we use the basic version of models without the GS component. The case of using two trajectories reduces to the pairwise-preference loss discussed in Section 2.

As shown in Fig. 2a and 2b, our generalized approach of using multiple trajectories to learn the reward function provides the flexibility to outperform the classical pairwise-preference learning. This is more apparent in the RewardMLE models, which are less sensitive to small errors in the ground-truth scores. In general, the optimal trajectory number may depend on the scoring quality.

**(b):** *Do different probabilistic transforms in reward learning objectives affect the performance?* We modify the basic version of the RewardNet and RewardMLE models in Table 1 by using the softmax transform and by using different powers in the escort transform in the reward learning losses Eqs. (2) and (3). For the escort transform, we consider $\Phi = (\cdot)^p, p \in \{1, 2, 3, 4\}$.

Figs. 2c and 2d plot the resulting Combined Scores. We see that the RewardMLE model is less sensitive to the choice of probabilistic transform — all the considered variants have a Combined Score of at least 104. In fact, changing its softmax transform used in Table 1 to the escort transform with power two improves the performance to 106.77. Thus, the choice of probabilistic transform provides an additional angle to improve the learned reward function and the entire ToD model.

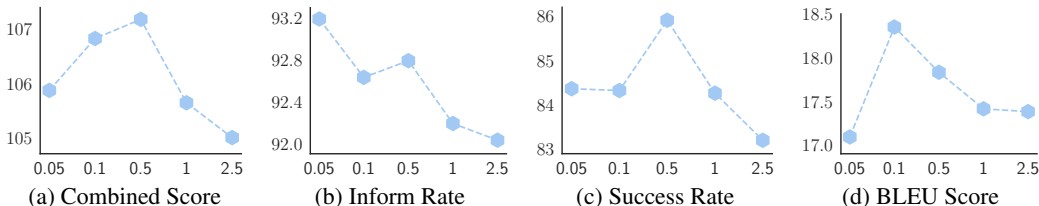

Figure 3: Line plots showing the four automatic evaluation metrics of the `RewardNet` + GS model in Table 1 under different $\alpha$ values in the generation-model loss Eq. (5). Results are the average over five seeds.

**(c):** *Is our method sensitive to the coefficient $\alpha$ in the generation-model loss Eq. (5)?* To investigate the robustness of our method under different weights for the policy-gradient optimization of the response-generation model. We select our best policy-gradient-based model in Table 1, the `RewardNet` + GS model, and vary the $\alpha$ coefficient in the generation-model loss Eq. (5). Fig. 3 plots the resulting four automatic evaluation metrics.

We see that our model is relatively robust to the choice of $\alpha$. The five variants in Fig. 3 all have Combined Scores of at least 105, higher than the best baseline result of 104.78 in Table 1. In fact, by changing the $\alpha$ coefficient to 0.5 from 0.1 used in Table 1, we achieve a *even better* Combined Score of $\approx 107.2$. Further, the capability of task completion and the fluency of the generated responses are both relatively insensitive to the choice of $\alpha$.

**(d):** *How does the addition of the policy-gradient method Gumbel-softmax help the performance?* Fig. 4 compares the performance of our models in Table 1, with error bars showing the standard deviation of the Combined Score over five seeds. It is clear that the addition of the Gumbel-softmax method can not only improve the score but also reduce the performance variation, which is apparent when comparing the `RewardMLE` model with the `RewardMLE` + GS model.

As discussed in Section 3.2, the Gumbel-softmax (GS) trick can be more advantageous than the classical REINFORCE method (Williams, 1992) for the policy-gradient update. As a demonstration, we conduct a toy experiment following Yin et al. (2019) and plot the results in Fig. 5. The task here is to learn the parameter $\psi$ of a $D$-dimensional categorical distribution to maximize a simple reward function. Specifically, denote the sigmoid function as $\sigma(\cdot)$, the goal is

$$\max_{\psi \in \mathbb{R}^D} \mathbb{E}_{x \sim \text{Cate}(\sigma(\psi))} \left[ f(x) \right], \quad f(x) \triangleq 0.5 + x/(D \cdot R), \ \forall x \in \{1, \ldots, D\},$$

where $\text{Cate}(\sigma(\psi))$ denotes the categorical distribution with probability vector $\sigma(\psi)$, and $D = R = 30$. The best $\sigma(\psi)$ is $(0, \ldots, 0, 1)$, leading to the optimal expected reward of $\approx 0.533$. We initialize $\psi = \mathbf{0}$ and use one sample for the stochastic gradient-ascent update, with a learning rate of 1.0.

The first row of Fig. 5 traces the objective function during the training process when using the true gradient, REINFORCE, and the GS for policy-gradient updates. We see that the REINFORCE method converges to a local maximum, while the GS method reaches the global optimum, as using the true gradient for updates. The second row shows the gradients for $\theta_1$ and $\theta_D$, where we see that gradient estimates from the REINFORCE method are both unstable and vanishing, compared to the GS method. The learned probabilities $\{\sigma(\psi)_1, \ldots, \sigma(\psi)_D\}$ is traced in the third row, where the red line is for $\sigma(\psi)_D$ that should ideally be 1, and the shadowed lines are for the other components that ought to be 0. The learning process of the GS method closely resembles that of using the true gradient, while REINFORCE oscillates around a local optimum. The last row of Fig. 5 plots the estimate of gradient variance via 500 samples, averaged over each component of the $\psi$ vector. The gradient variance of the REINFORCE method is on the order of $10^{-2}$ at the beginning and converges to roughly $10^{-4}$, while the GS is $10^{-6}$ throughout the training process. This toy experiment shows that a low-variance method, such as the GS, can be critical to the success of policy-gradient training.

## 5.3 FURTHER ANALYSIS

**Human evaluation.** For a more comprehensive evaluation of our method, we conduct a human evaluation on the quality of the generated responses, where our model and the top two baselines in Table 1, GPT-Critic and CASPI, are compared. We follow the evaluation protocol in prior work (*e.g.*, Zhang et al., 2020; Ramachandran et al., 2021; Jang et al., 2022) to evaluate on two metrics: 1) **Appropriateness**: measures the appropriateness of the generated response under the context of the dialogue turn; 2) **Fluency**: evaluates the comprehensibility and coherence of the generated response. We randomly picked 50 turns in the test set and showed to 10 evaluators the responses

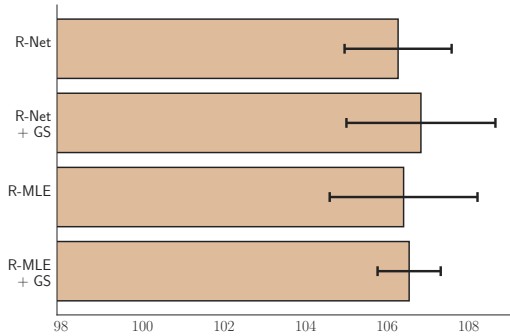

Figure 4: Bar plot comparing our four models in Table 1. Mean and one standard deviation over five random seeds are shown. "R-Net" denotes RewardNet. "R-MLE" is RewardMLE. "GS" is Gumbel-softmax.

Figure 5: Toy experiment comparing REINFORCE and Gumbel-softmax in reward maximization, the estimated gradients, the estimated probabilities, and the gradient variance. See Section 5.2 for details.

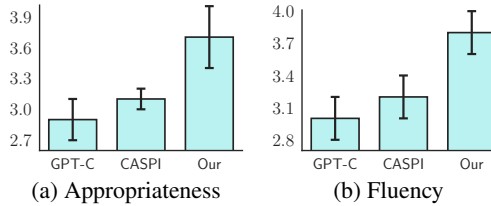

| (a) Appropriateness | (b) Fluency |
| --- | --- |

Figure 6: Bar plots for the results of human evaluation on appropriateness and fluency, showing the mean and one standard deviation of each method. The scores are on a 5 scale and higher scores indicate better results. "GPT-C" denotes GPT-Critic. Details for the setup of human evaluation are discussed in Section 5.3.

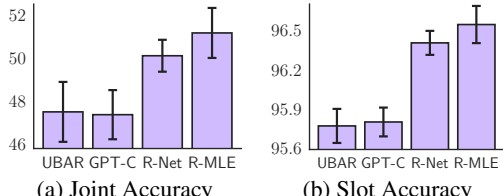

| (a) Joint Accuracy | (b) Slot Accuracy |
| --- | --- |

Figure 7: Bar plots for the quality of the generated dialogue states, showing the mean and standard deviation of two metrics. "GPT-C" denotes GPT-Critic, "R-Net" denotes RewardNet, "R-MLE" is RewardMLE. The results of UBAR and GPT-Critic are from Jang et al. (2022). Our results are over five random seeds.

generated from each method, together with the dialogue history up to that turn. The method names were anonymized. The evaluators were asked to read the dialogue history and score the response on a 5-Point Likert Scale $\{1, 2, 3, 4, 5\}$, where score 5 is the highest and 1 the lowest.

Fig. 6 summarizes the evaluation results. We see that our model outperforms the baselines in both the appropriateness and fluency scores. The human-evaluation results coincide with our comparatively good dialogue-task completion and BLEU score in Table 1.

**Examples of the generated dialogues.** Tables 3 and 4 in Appendix A conduct two case studies comparing the generated responses from our method with those from the baselines GPT-Critic and CASPI. We additionally annotate the generated responses to discuss the quality of those generations. These examples show that the responses from our model compare favorably with the baselines in both task completion and comprehensibility, aligning with the automatic and human evaluations.

**Quality of the DST.** To further understand the performance gain of our models, we compare our basic RewardNet and RewardMLE models in Table 1 with the baselines UBAR and GPT-Critic on the quality of the generated dialogue states. Fig. 7 plots the results of the dialogue state prediction, measured by the two metrics Joint (Goal) Accuracy and Slot Accuracy (Wu et al., 2019a). We see that our two models have more accurate DST than the two baselines, which can be related to their better performance in Table 1. Interestingly, the DST of the RewardMLE model is also better than that of the RewardNet model. This may suggest that a better reward model not only benefits the learning of response generation, but also the DST. These two losses are jointly minimized in training the ToD model, and thus a good response-generation loss from a better reward model may help the optimization of the DST loss.

## 6 CONCLUSION

In this paper, we aim to answer the question of how to efficiently learn and utilize a reward function for training the E2E ToD agents. We answer this question by introducing two generalized reward-learning objectives, and utilize a stable policy-gradient method to guide the training of the E2E ToD agents. Future work includes extending our reward-learning objectives to other applications, such as the question-answering with verification.

ACKNOWLEDGMENTS

S. Yang and M. Zhou acknowledge the support of NSF-IIS 2212418 and the Texas Advanced Computing Center (TACC) for providing HPC resources that have contributed to the research results reported within this paper.

ETHICAL STATEMENT

We develop our methods based on the publicly available MultiWOZ 2.0 dataset (Budzianowski et al., 2018). It is important to note that, like other ToD models, our implementation will likely reflect the socio-economic and entity biases inherent in the MultiWOZ dataset (Qian et al., 2021). We initialize model parameters from the pre-trained BART model. And a comprehensive analysis of certain biases captured by BART is outside our scope. Besides, we invited volunteers for the human evaluation, with transparent and detailed explanations and communications on data usage, research intent, occupied hours, *etc.*

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

# Appendix

## A  EXAMPLES OF THE GENERATED DIALOGUES

Tables 3 and 4 show two case study comparing the generated responses from our method and from the baselines GPT-Critic and CASPI. Our method outperforms the baselines in terms of both task completion and the quality of the generated expressions.

Table 3: Generated and original dialogue for ID PMUL4610. "User" denotes User Utterance. "System" denotes true System Response. Red annotations explain unsatisfactory responses and green for satisfactory ones.

| | | PMUL4610 |
|---|---|---|
| Turn Num. | Method | Original / Generated Dialogue |
| 1 | User | i am looking for a museum in the east to visit. |
| | GPT-Critic | there are [value_choice] [value_type] in the [value_area]. [value_name] is [value_price]. [value_name] is [value_price].                            // Do not make recommendation. |
| | CASPI | there are [value_choice] [value_type] in the [value_area]. i would recommend [value_name].                            // Do not explain recommendation. |
| | Ours | there are [value_choice] [value_type] in the [value_area]. i recommend [value_name] because it is [value_price] to get in .            // Make recommendation and explain. |
| | System | we have [value_choice] [value_type] in the [value_area] area , i just went to the [value_name] and they have new exhibits . would you like more information on them ? |
| 2 | User Utterance | sounds wonderful! can you give me their postcode? |
| | GPT-Critic | their postcode is [value_postcode]. is there anything else i can help you with? |
| | CASPI | sure thing ! the postcode is [value_postcode]                            // Do not elicit conversation.. |
| | Ours | sure , the postcode is [value_postcode]. is there anything else i can help you with ? |
| | System Response | their postcode is [value_postcode]. can i help you with anything else today ? |
| 3 | User Utterance | i am also looking for a restaurant in the east part of town in the moderate price range. |
| | GPT-Critic | there are [value_choice] restaurant -s in that area. [value_name] and [value_name].                            // Do not elicit conversation |
| | CASPI | [value_name] is a [value_price] restaurant in the [value_area].      // Do not elicit conversation. |
| | Ours | there are [value_choice] restaurant -s that fit your criteria . [value_name] serves [value_food] food and is located in the [value_area] part of town . would you like their address ? |
| | System Response | there are [value_choice] [value_food] restaurant -s and [value_choice] [value_food]. which would you prefer ? |
| 4 | User Utterance | give me the address of the 1 you recommend |
| | GPT-Critic | [value_name] is located at [value_address]. is there anything else i can help you with?                            // Do not make recommendation. |
| | CASPI | [value_name] is a favorite of mine . they are located at [value_address]                            // Do not elicit conversation.                            . |
| | Ours | [value_name] is a great [value_food] restaurant in the [value_area]. // More plural expression. the address is [value_address].                            // Do not elicit conversation. |
| | System Response | i recommend [value_name], they are t [value_address], postal is [value_postcode], can i assist with anything else ? |
| 5 | User Utterance | not at this time, thank you. |
| | GPT-Critic | thank you for using our system. goodbye |
| | CASPI | you are welcome . have a great day ! |
| | Ours | you are welcome . have a great day ! |
| | System Response | thank you for calling. please let us know if we can assist you in the future. goodbye. |

Table 4: Generated and original dialogue for ID SNG1012. "User" denotes User Utterance. "System" denotes true System Response. Red annotations explain unsatisfactory responses and green for satisfactory ones.

| | | SNG1012 |
|---|---|---|
| Turn Num. | Method | Original / Generated Dialogue |
| 1 | User | i need to find a guest house with a 3 star rating |
| | GPT-Critic | i have [value_choice] options for you. do you have a price range or area of town you would like to stay in? |
| | CASPI | i have [value_choice] [value_type] that meet your criteria . do you have a preference of area or price range ? |
| | Ours | i have [value_choice] [value_type] with [value_stars] stars . would you prefer a [value_price] or [value_price] price range? // Do not ask location. |
| | System | we have [value_choice] such [value_type]. do you have a preferred location? |
| 2 | User | no, i don't. i want 1 that include -s free wifi. |
| | GPT-Critic | [value_name] is a [value_price] [value_stars] star [value_type] in the [value_area] with free wifi and parking. would you like me to book it for you? |
| | CASPI | i have [value_choice] [value_type] that meet your criteria . [value_name] and [value_name]. both are [value_price] and offer free wifi and parking . do you have a preference ? |
| | Ours | i have [value_choice] [value_type] that have free wifi . i would recommend the [value_name]. would you like me to book a room ? |
| | System | [value_name], [value_name], and [value_name] are all available if you would like 1 of those? |
| 3 | User | please book the hamilton lodge, for 6 people. |
| | GPT-Critic | booking was successful. reference number is : [value_reference]. is there anything else i can help you with?        // Finish booking before acquire sufficient information. |
| | CASPI | i can book the [value_name] for you . how many nights will you be staying ? |
| | Ours | i would be happy to book the [value_name] for you . what day would you like to arrive ? |
| | System | i would be happy to. how many nights would you like to stay? and i 'll need to know the day you would like to start your stay. |
| 4 | User | i 'll be staying 2 nights starting on wednesday. oh, and there are 6 of us. |
| | GPT-Critic | i have booked you a room at [value_name] for [value_people] people for [value_stay] nights starting [value_day]. your reference number is [value_reference]. is there anything else i can help you with?    // Dialogue diverges from ground truth. |
| | CASPI | i am sorry , but there are no rooms available for those dates . would you like to try a different day or a shorter stay ? |
| | Ours | i am sorry , but i was unable to book that [value_type]. would you like to try another 1 ? |
| | System | i am sorry, my reservation attempt was unsuccessful. will another day or length of stay work? |
| 5 | User | can you try the same parameters at another hotel? |
| | GPT-Critic | i have booked you at [value_name] for [value_stay] nights starting [value_day] for [value_people] people. your reference number is [value_reference]. is there anything else i can help you with?     // Repeating diverged dialogue. |
| | CASPI | i am sorry , i am not able to book at another [value_type]      // Fail to continue conversation.. |
| | Ours | i have [value_name] in the [value_area]. it is a [value_price] [value_stars] star [value_type]. would you like me to book that for you ? |
| | System | of course, could you tell me the area you would like to stay in so i can narrow it for you? |
| 6 | User | the are does not matter, just 3 stars and wifi at a guest house please. |
| | GPT-Critic | would you like me to book it for you?                              // Fail to complete booking. |
| | CASPI | i was able to book you at [value_name]. reference number [value_reference]. |
| | Ours | i was able to book you at [value_name]. your reference number is [value_reference]. is there anything else i can help you with ? // More plural expression and elicit conversation. |
| | System | i have got you a reservation for [value_people] at hobson's house for [value_stay] nights. your reference number is [value_reference]. |
| 7 | User | thank you so much! that should be all i need. |
| | GPT-Critic | you are welcome. have a great day. |
| | CASPI | you are welcome. have a great day! |
| | Ours | you are welcome. have a great day! |
| | System | you are welcome. enjoy your day. |

# B    ALGORITHMIC DETAILS

**Prepossessing.**    The raw corpus is prepossessed by common practice in the ToD literature. Specifically, we represent the database (DB) query results as one-hot vectors following Budzianowski et al. (2018), use domain-adaptive delexicalization proposed by Wen et al. (2016), and generate delexicalized responses with placeholders for specific DST/DB information as in Zhang et al. (2020).

**Implementation of the response model.**    Our model in Section 5 is based on the MinTL ToD model (Lin et al., 2020), which uses the pre-trained BART-large model (Lewis et al., 2019). MinTL directly works on the system response and does not explicitly output the dialogue act. Our proposed method in Section 3 is applied to the response training, and we retain the DST-training loss in MinTL. Our model is trained by fine-tuning BART on the training set and early-stopping by the validation set.

**Implementation of the reward model.**    Our reward model is implemented by the encoder part of the BART-base model, followed by a simple two-layer MLP. The output of the reward model is scaled to $[0, 1]$ via the sigmoid function. The input to the reward model is the concatenation of the belief state, system response, and dialogue goal, at each turn of the sampled dialogue rollout. The model outputs the reward of each turn in the dialogue rollout, which is summed and fed into the losses proposed in Section 3.1. We use the HuggingFace library (Wolf et al., 2019) to implement our reward model.

Algorithm 1 illustrates the pipeline of our methods.

---

**Algorithm 1** Pipeline of the proposed reward learning and utilization methods for training E2E ToD agents.

---

**Input:** Reward function $\mathcal{R}_\theta(o, a, g)$, ToD agent $\pi_\phi$, dataset $\hat{D} := \left\{ \left( g^{(k)}, (o_t^{(k)}, a_t^{(k)})_{t=0}^T \right) \right\}_{k=1}^K$, number of iterations $M_1$ and $M_2$, probabilistic transform function $\Phi$, hyperparameters $N$, $\alpha$.

**for** iteration $\in \{1, \ldots, M_1\}$ **do**
    Sample $N$ dialogue trajectories from the dataset $\hat{D}$.
    Optimize $\mathcal{R}_\theta$ via `RewardNet` (Eq. (2)) or `RewardMLE` (Eq. (3)).
**end for**
Fix the reward function $\mathcal{R}_\theta$.
**for** iteration $\in \{1, \ldots, M_2\}$ **do**
    Sample a batch of transition tuples $\left( g^{(k)}, (o_t^{(k)}, a_t^{(k)}) \right)$ from the dataset $\hat{D}$.
    Optimize the ToD agent $\pi_\phi$ via Eq. (6).
**end for**

**Output:** Trained ToD agent $\pi_\phi$.

---

## C  COMPARISON WITH SOME OTHER REWARD-LEARNING METHODS IN RL-BASED DIALOGUE AGENTS

As an additional discussion on related work, in this section we briefly compare our work with three other reward-learning methods in RL-based dialogue agents, *i.e.*, Saito (2018), Hu et al. (2018), and Li et al. (2020).

These three papers all focus on the dialogue-management module of the pipeline design, wherein the action spaces of the agents are some abstract dialogue acts rather than the human-language-like system response as in our paper. The possible system responses are fixed in these three papers. For example, Hu et al. (2018) train the agent to select from "the set of the indices to all available questions in the Q20 game;" and Li et al. (2020) have an action space of size 300. As discussed in Section 1, such a pipeline approach requires intensive structural designs, such as determining the possible questions in the Q20 games; and may not enjoy the language plurality and conversation elicitation that our E2E model could offer, *e.g.*, as shown in Table 4. Due to the complexity and the much higher dimension of our action space as the system responses, the methods proposed in these three papers are not directly applicable to our setting, which will be discussed in detail below.

The method proposed in Saito (2018) require specially-designed curriculum data and hand-crafted decomposition of the entire task into sub-tasks, which are not readily available and are non-trival for large-scale multi-domain dialogue corpus such as our tested MultiWOZ 2.0 dataset. The use of progressive neural networks to provide reward information in Saito (2018) require additional computation and memory complexity and thus may not scale to transformers. Meanwhile, our method scales well to transformers, as shown in our experiments (Section 5). Further, the method in Saito (2018) may only be feasible on the task of constrained information-retrieval, but not on some more general tasks such as the booking requirement in the tested MultiWOZ 2.0 dataset. Our experimental results show that our method is capable of such tasks.

Similar to our work, Hu et al. (2018) propose a neural network to approximate the reward function to deliver immediate rewards at each timestep. Apart from the aforementioned simple action space, Hu et al. (2018) use the long-term return $G_t$ as a surrogate indicator of $r_{t+1}$ to train the reward function (Eq. (6) of Hu et al. (2018)), which is lack of justification. By contrast, as discussed in Section 2 and Section 3, our method is based on the classical approaches in the learning-to-rank (LTR) literature and extends the classical reward-learning-from-preferences into utilizing multiple dialogue trajectories simultaneously to optimize the reward function.

As discussed before, Li et al. (2020) consider a relatively small action space of size 300 and learns the reward model via the GAN structure, which may not stably scale up to high-dimensional action space such as the system response in our E2E ToD system. The learned reward function in Li et al. (2020) only measure the probability that the input is from the real-data distribution, *i.e.*, only considers *a pair of* dialogue state $s_t$ and the corresponding system action $a_t$. This reward function does not consider the success of the entire dialogue, which is intuitively less favorable to the E2E ToD systems. By contrast, our method trains a reward function that aligns with some evaluations on the *entire* dialogue trajectories, which is more directly related to the usage of the ToD systems.

We further note that apart from the reward-learning method, our paper also discusses using the Gumbel-softmax trick as a more stable method to train the E2E ToD systems, and conducts a toy experiment in Section 5.2 to illustrate the advantage of the Gumbel-softmax trick over the classical REINFORCE method. This is not covered in the three prior works Saito (2018), Hu et al. (2018), and Li et al. (2020).

Finally, these three prior works use online RL methods such as DQN, REINFORCE, and PPO, which require environmental interactions and are thus less practical, as discussed in Section 1. In contrast, our method allows training E2E response-generation models from static datasets by utilizing offline RL techniques (*e.g.*, Levine et al., 2020; Fujimoto & Gu, 2021; Yang et al., 2022a;b;c).

# D   DETAILED COMPARISON WITH CASPI

Table 5 further compares CASPI and our two methods: `RewardNet` +GS, $N = 3, \Phi = (\cdot)^1$ and `RewardMLE` +GS, $N = 5, \Phi = \exp(\cdot)$, showing a detailed breakdown of the scores onto each of the five tested random seeds.

The performances of both our methods and CASPI are relatively stable across random seeds. In particular, both of our methods have higher Combined Score than CASPI *on each of the five tested random seeds*. This stable improvement of our methods over CASPI aligns with our intuition discussed in Section 3 and our main experimental results in Section 5.1.

We note that both the Average score and the Median score across the tested random seeds are valid metrics for performance comparison. Nevertheless, there is some ambiguity in calculating the "Median" of the Combined Score, namely, should it be the median of the Combined Scores on each random seed, or should it be calculated as $\mathrm{Median(Combined\ Score)} \triangleq (\mathrm{Median(Inform)} + \mathrm{Median(Success)}) \times 0.5 + \mathrm{Median(BLEU)}$? The first way aligns better with the definition of "Median" while the second way aligns better with the definition of Combined Score. Such an ambiguity is cleared out when using the Average as the evaluation metric. Besides, the metric Average score is widely used in prior work (*e.g.*, Zhang et al., 2020; Lin et al., 2020; Jang et al., 2022). With these considerations, we choose to report the Average over five random seeds in our experimental section (Section 5).

Table 5: Per random-seed results of the E2E response generation task on the MultiWOZ 2.0 dataset, comparing CASPI and our two models: `RewardNet` +GS, $N = 3, \Phi = (\cdot)^1$ and `RewardMLE` +GS, $N = 5, \Phi = \exp(\cdot)$. Here, $(\cdot)^1$ denotes the power function with power 1. "Comb." is the Combined Score. The row "Median" shows the median score of the corresponding column over the five random seeds.

| Seed | CASPI | | | | `RewardNet` +GS, $N = 3, \Phi = (\cdot)^1$ | | | | `RewardMLE` +GS, $N = 5, \Phi = \exp(\cdot)$ | | | |
|---|---|---|---|---|---|---|---|---|---|---|---|---|
| | Inform | Success | BLEU | Comb. | Inform | Success | BLEU | Comb. | Inform | Success | BLEU | Comb. |
| 111 | 88.19 | 80.88 | 18.88 | 103.42 | 90.49 | 82.68 | 18.42 | 105.01 | 93.99 | 84.18 | 16.82 | 105.91 |
| 333 | 91.69 | 83.58 | 18.17 | 105.81 | 94.39 | 85.09 | 18.69 | 108.43 | 95.10 | 85.09 | 17.51 | 107.61 |
| 555 | 91.99 | 81.78 | 17.21 | 104.10 | 91.29 | 82.18 | 18.54 | 105.28 | 91.99 | 83.68 | 18.67 | 106.51 |
| 777 | 93.39 | 83.48 | 17.13 | 105.57 | 91.89 | 84.18 | 18.36 | 106.40 | 92.79 | 83.18 | 17.73 | 105.72 |
| 999 | 91.59 | 84.28 | 17.09 | 105.03 | 95.10 | 87.49 | 17.74 | 109.04 | 91.59 | 83.38 | 19.48 | 106.97 |
| Average | 91.37 | 82.80 | 17.70 | 104.78 | 92.63 | 84.32 | 18.35 | 106.83 | 93.09 | 83.90 | 18.04 | 106.54 |
| Median | 91.69 | 83.48 | 17.21 | 105.03 | 91.89 | 84.18 | 18.42 | 106.40 | 92.79 | 83.68 | 17.73 | 106.51 |

# E   EXPERIMENTS WITH THE GALAXY

To further demonstrate the efficacy and applicability of our approach, we apply our reward learning and utilization methods to the recently proposed GALAXY backbone (He et al., 2022), which achieves competitive performance on the E2E response-generation task on the MultiWOZ 2.0 dataset. We note that the GALAXY paper does not disclose *how many* and *which* random seeds were used to obtain its main results; and the official codebase fixes the random seed as 10. This makes us unsure if its reported results are only from this single seed of 10. To mitigate the randomness in the optimization process, we re-run the vanilla GALAXY on the five random seeds used to generate our main results and compare it on these seeds with the variants equipped with our methods `RewardNet` +GS, $N = 3, \Phi = (\cdot)^1$ and `RewardMLE` +GS, $N = 5, \Phi = \exp(\cdot)$. Table 6 shows a detailed breakdown of the scores on each of the five random seeds. Note that since we use different random seeds than the original GALAXY paper, we are unable to get its reported scores.

We see from Table 6 that adding our reward learning and utilization methods improves the performance of the vanilla GALAXY, in almost all evaluation metrics, in both the Average and the Median scores. In particular, adding our `RewardMLE` +GS method improves the average Combined Score of the vanilla GALAXY by 3.27, and adding our `RewardNet` +GS improves the vanilla GALAXY by 2.11. These relatively significant improvements may further demonstrate the effectiveness and general applicability of our proposed methods.

Table 6: Per random-seed results of the E2E response-generation task on the MultiWOZ 2.0 dataset, comparing the vanilla GALAXY and the variants with our proposed methods: `RewardNet` +GS, $N = 3, \Phi = (\cdot)^1$ and `RewardMLE` +GS, $N = 5, \Phi = \exp(\cdot)$. Here, $(\cdot)^1$ denotes the power function with power 1. "Comb." is the Combined Score. The row "Median" shows the median score of the corresponding column over the five tested random seeds.

| Seed | GALAXY | | | | `RewardNet` +GS, $N = 3, \Phi = (\cdot)^1$ | | | | `RewardMLE` +GS, $N = 5, \Phi = \exp(\cdot)$ | | | |
| | Inform | Success | BLEU | Comb. | Inform | Success | BLEU | Comb. | Inform | Success | BLEU | Comb. |
| --- | --- | --- | --- | --- | --- | --- | --- | --- | --- | --- | --- | --- |
| 111 | 93.60 | 85.40 | 19.97 | 109.47 | 91.30 | 83.00 | 19.03 | 106.18 | 92.50 | 84.10 | 18.89 | 107.19 |
| 333 | 86.50 | 77.90 | 17.87 | 100.07 | 90.00 | 82.00 | 17.95 | 103.95 | 90.90 | 82.70 | 18.27 | 105.07 |
| 555 | 89.20 | 81.60 | 19.96 | 105.36 | 92.00 | 83.40 | 19.44 | 107.14 | 93.90 | 85.50 | 19.47 | 109.17 |
| 777 | 90.40 | 81.90 | 18.81 | 104.96 | 93.70 | 85.10 | 18.90 | 108.30 | 95.80 | 86.10 | 19.36 | 110.31 |
| 999 | 88.30 | 79.40 | 18.59 | 102.44 | 92.60 | 83.60 | 19.19 | 107.29 | 91.90 | 82.50 | 19.70 | 106.90 |
| Average | 89.60 | 81.24 | 19.04 | 104.46 | 91.92 | 83.42 | 18.90 | 106.57 | 93.00 | 84.18 | 19.14 | 107.73 |
| Median | 89.20 | 81.60 | 18.81 | 104.96 | 92.00 | 83.40 | 19.03 | 107.14 | 92.50 | 84.10 | 19.36 | 107.19 |

# F    EXPERIMENTS ON THE MULTIWOZ 2.1 DATASET

To test the efficacy of our proposed methods on additional datasets, we run our two methods `RewardNet` + GS, $N = 3, \Phi = (\cdot)^1$ and `RewardMLE` + GS, $N = 5, \Phi = \exp(\cdot)$ on the MultiWOZ 2.1 dataset. Table 7 compares our two methods with the baselines SimpleTOD and UBAR in the main evaluation (Table 1), which also report results on the MultiWOZ 2.1 dataset. Additionally, we also present our rerun of CASPI on this dataset. As in Table 1, the Combined Scores of our methods are generally better than the baselines. In fact, our methods achieve both good task completion (Inform and Success rates) and fluent generated responses (BLEU score).

Table 8 shows a detailed breakdown of the scores of CASPI and our two methods on each of the five tested random seeds. We see that the scores of our methods are generally robust across random seeds. In fact, on four out of those five seeds, at least one of our methods performs better than CASPI. Further, our methods have higher Average and Median scores than CASPI on each of the four evaluation metrics. This set of experiments may further demonstrate the efficacy of our proposed methods.

Table 7: Results of the E2E response-generation task on the MultiWOZ 2.1 dataset. The best result on each metric is bold. The results of SimpleTOD and UBAR are from the original paper. The results of CASPI are from our reproduction. All our provided results are the average over five random seeds. "GS" denotes the Gumbel-softmax trick. $(\cdot)^1$ denotes the power function with power 1.

| Algorithms | Inform | Success | BLEU | Combined Score |
| --- | --- | --- | --- | --- |
| SimpleTOD (Hosseini-Asl et al., 2020) | 85.00 | 70.50 | 15.23 | 92.98 |
| UBAR (Yang et al., 2021) | **95.70** | 81.80 | 16.50 | 105.25 |
| CASPI (Ramachandran et al., 2021) | 91.43 | 83.50 | 17.93 | 105.40 |
| `RewardNet` + GS, $N = 3, \Phi = (\cdot)^1$ | 92.79 | **84.48** | 17.99 | 106.62 |
| `RewardMLE` + GS, $N = 5, \Phi = \exp(\cdot)$ | 92.87 | 83.90 | **18.73** | **107.11** |

Table 8: Per random-seed results of the E2E response-generation task on the MultiWOZ 2.1 dataset, comparing the CASPI and the variants with our proposed methods: `RewardNet` +GS, $N = 3, \Phi = (\cdot)^1$ and `RewardMLE` +GS, $N = 5, \Phi = \exp(\cdot)$. Here, $(\cdot)^1$ denotes the power function with power 1. "Comb." is the Combined Score. The row "Median" shows the median score of the corresponding column over the five tested random seeds.

| Seed | CASPI | | | | `RewardNet` +GS, $N = 3, \Phi = (\cdot)^1$ | | | | `RewardMLE` +GS, $N = 5, \Phi = \exp(\cdot)$ | | | |
| | Inform | Success | BLEU | Comb. | Inform | Success | BLEU | Comb. | Inform | Success | BLEU | Comb. |
| --- | --- | --- | --- | --- | --- | --- | --- | --- | --- | --- | --- | --- |
| 111 | 89.39 | 82.08 | 18.14 | 103.88 | 96.6 | 87.19 | 17.23 | 109.13 | 93.39 | 84.58 | 19.39 | 108.38 |
| 333 | 90.99 | 82.08 | 18.25 | 104.79 | 91.69 | 83.78 | 18.43 | 106.17 | 92.19 | 81.78 | 17.36 | 104.35 |
| 555 | 91.59 | 82.68 | 17.9 | 105.04 | 91.79 | 85.29 | 17.63 | 106.17 | 93.19 | 85.49 | 19.08 | 108.42 |
| 777 | 91.89 | 84.28 | 17.78 | 105.87 | 91.49 | 82.68 | 18.23 | 105.32 | 93.29 | 83.78 | 19.15 | 107.74 |
| 999 | 93.29 | 86.39 | 17.6 | 107.44 | 92.39 | 83.48 | 18.41 | 106.35 | 92.29 | 83.78 | 18.65 | 106.69 |
| Average | 91.43 | 83.50 | 17.93 | 105.40 | 92.79 | 84.48 | 17.99 | 106.62 | 92.87 | 83.90 | 18.73 | 107.11 |
| Median | 91.59 | 82.68 | 17.90 | 105.04 | 91.79 | 83.78 | 18.23 | 106.17 | 93.19 | 83.88 | 19.08 | 107.74 |

# G IMPLEMENTATION DETAILS

Our implementation is based on the official codebase of MinTL and CASPI. Apart from the hyper-parameters discussed in Section 5.2, most other hyperparameters and the training procedure of our models follow MinTL and CASPI. In addition to the discussion of our method in Section 3, we list the important hyperparameters for training our reward model in Table 9 and the important hyperparameters for training our response-generation model in Table 10. Both BART models use the default token length of 142.

We note that unlike CASPI which uses the dialogue acts as the actions, the action space of our reward model is the system response, which is a combinatorial space of the vocabulary. We use this action space because we want to pass gradients from the reward model to the E2E response-generation model during the training process, where the output of the E2E model is the human-language-like system response.

For the reward-function learning, we do not change the length of the trajectories in the dataset. The reward model is updated by the preference scores/orderings among multiple trajectories of the same length. Our intuition is that trajectories of the same length may roughly correspond to tasks of similar complexity, making the preference among them more comparable and meaningful. This approach is also taken by the prior work CASPI.

Our tested MultiWOZ2.0 dataset is publicly available at https://github.com/budzianowski/multiwoz, and the MultiWOZ2.1 dataset is available at https://github.com/thu-coai/ConvLab-2/tree/master/data/multiwoz.

Table 9: Hyperparameters for training our reward model.

| Hyperparameter | Value |
|---|---|
| Action space | System response |
| Gradient clipping norm | 1.0 |
| Learning rate | 3e-5 |
| Gradient accumulation steps | 16 |
| Batch size | 4 |
| Context window | 2 |
| Learning-rate decay | 0.8 |
| Learning-rate scheduler | ReduceLROnPlateau |
| Scheduler patience | 3 |
| Early-stop count | 7 |
| Backbone | BART-base |

Table 10: Hyperparameters for training our response generation model.

| Hyperparameter | Value |
|---|---|
| Action space | System response |
| Gradient clipping norm | 1.0 |
| Learning rate | 3e-5 |
| Gradient accumulation steps | 8 |
| Batch size | 8 |
| Context window | 2 |
| Learning-rate decay | 0.8 |
| Learning-rate scheduler | LambdaLR |
| Early-stop count | 7 |
| Seed | {111, 333, 555, 777, 999} |
| Backbone | BART-large-cnn |

