# OpenReview forum: "Fantastic Rewards and How to Tame Them: A Case Study on Reward Learning for Task-oriented Dialogue Systems"
_ICLR.cc/2023/Conference — ICLR 2023 poster_

### Official Review · Reviewer_NRwE · 2022-10-21

**Confidence:** 4
**Correctness:** 3
**Technical Novelty And Significance:** 3
**Empirical Novelty And Significance:** 3
**Recommendation:** 6

**Clarity, Quality, Novelty And Reproducibility:**

## clarity
The goal and structure of this paper are clear and easy to follow. In section 5, the author discusses details of their methods and analysis them with toy examples. Section 5.2 (a) discusses the number of trajectories; I do not understand this sentence ‘We hypothesize that the optimal trajectory number depends on the scoring quality.’  Does there have theoretical reasons to prove that more than two trajectories can bring more advantages than the classical pairwise preference? I wonder if this N-trajectories idea can apply to other scenarios since most previous works use pairwise trajectories.

## quality
Quality overall is good except for the mentioned issues with respect to reproducibility below.

## novelty
it is a combination of existing techniques, but the combination itself is novel

## reproducibility
Code is not provided as far as I can see. Regarding the reproduction of CASPI, it is not clear which version from the official CASPI codebase did the author run to get the results in Table 1. It should be possible to show the Median score of the RewardNet and RewardMLE experiments and compare them to CASPI original paper. It is not mentioned how the authors choose the length of the trajectories to update the reward model. If all of the dialogues have an equal length of trajectory, could it happen that the agent arrives at the goal earlier than the set length or need more steps to achieve it? (e.g., Simple dialogue vs. Complex dialogue)

**Strength And Weaknesses:**

## Strengths
- The goal and structure of this paper are clear and easy to follow
- This paper shows strong experiment results. Their methods achieve the best results on the MultiWOZ 2.0 dataset.
- reasonable combination of well-known RL methods, new research direction in TOD domain

## Weaknesses
-  The novelty of reward techniques is limited
- no implementation available, reproducibility unclear

**Summary Of The Paper:**

This paper introduces two variants of reward learning functions in RL-based task-oriented dialogue (TOD) to address the issues of sparse and delayed rewards. RewardNet optimizes the rewards by computing each trajectory's total automatic evaluation scores and then sorting all trajectories by the scores. The RewardMLE uses only the ranking order instead of the automated evaluation scores. The ranked N dialogue trajectories are inputs into a neural network to learn the new rewards.
The RewardNet is updated by optimizing the cross entropy between accumulated predicted rewards and automatic evaluation scores. The negative log-likelihood is used for the RewardMLE.
The policy gradient-based agent predicts the action and updates its parameters with the learned rewards. Experiments  are performed on the MultiWOZ 2.0 dataset.

**Summary Of The Review:**

This paper applied the learning-to-rank-based rewards functions to TOD tasks. The basic idea behind RewardNet is similar to the method proposed by the CASPI paper. Additionally, this paper runs more experiments with different numbers of trajectories, introduces a new variant, RewardMLE, and discusses the policy gradient-based agent with the Gumbel-softmax trick. The novelty of their main reward techniques is limited. Still, this paper proposes a reasonable combination of well-known RL methods and a meaningful research direction with strong experiment results for the TOD domain. It’s useful for future work on the MultiWOZ 2.0 dataset to compare.  The author didn’t provide their implementations; I’m unsure about the reproducibility.



Minor suggestions: be sure to provide the full term when introducing an acronym; for example, section 3.2 (GS) occurs before its acronym Gumbel-softmax(GS).  It would be evident if Figure 1describe more details about the framework.

---

> ### Author Response · Authors · 2022-11-13
> **Response to Reviewer NRwE (Part 1 / 3)**
>
> We appreciate Reviewer NRwE for your insightful comments.
> Please would you check our general response regarding the novelty and the reproducibility of our work.
> Below are our detailed responses to your concerns.
>
>
> >**Q1:**   I do not understand this sentence ‘We hypothesize that the optimal trajectory number depends on the scoring quality.’
>
> **A:**
> Sorry for the confusion. This sentence is meant to hypothesize that if the trajectory ordering in the dataset is noisy and does not well reflect the true underlying preference ordering, then training the reward model simultaneously using more trajectories may not always be beneficial.
>
> Our intuition is based on the following thought experiment. Suppose the true preference ordering is $\tau_1 \succ \tau_2 \succ \tau_3$. And suppose the trajectory ordering within the dataset is a noisy version of the truth, being $\tau_1 \succ \tau_3 \succ \tau_2$. If we use the RewardMLE objective with the number of trajectories equals three, we will fit our reward model onto the wrong ordering $\tau_1 \succ \tau_3 \succ \tau_2$. By contrast, if we use the RewardMLE objective with the number of trajectories equals two, we will fit our reward model onto three ordering $\tau_1 \succ \tau_3$ (correct), $\tau_1 \succ \tau_2$ (correct), and $\tau_3 \succ \tau_2$ (wrong), of which two are correct and one is wrong.
> In this case, using two trajectories can be intuitively better than using three trajectories.
> Similar thought experiment can also be constructed for the RewardNet objective.
> Therefore, we provide the referred sentence as a caveat that, in practice, the number of trajectories used for the reward-learning losses depends on how faithful the trajectory ordering in the dataset reflects the true preference ordering.
>
> Nevertheless, as shown in our experiments, using the Combined Score to generate the trajectory ordering does not suffer much from the noisy-ordering issue.
> Especially for the RewardMLE models that are less sensitive to small perturbations in the evaluation scores, as shown on Fig. 2b, using multiple trajectories to learn the reward function generally outperforms the classical pairwise approach.
>
>
> >**Q2:**  Reasons to show that more than two trajectories can bring more advantage than the classical pairwise preference?
>
> **A:**  As mentioned at the beginning of Section 3.1, the main advantage of using multiple trajectories is the efficiency.
>
> As a simple thought experiment, suppose we have four trajectories to learn a simple reward function (e.g., tabular reward function), for which the optimization can be conducted exactly, If we use all four trajectories to learn the reward function, we only need one reward-function update to get a reward function that reflects the desired ordering of the trajectories.
> By contrast, if we use pairwise trajectory-preference to learn the reward function, we need at least three updates to get the desired reward function.
>
> In practice, this advantage can be more significant when we have a large number of trajectories and when we train the reward model using a fixed amount of stochastic gradient descent updates.
> In this scenario, our approach of using multiple trajectories provides more information for reward-function learning at each optimization step.
> This may help learning a more effective reward function within the optimization budget, compared with the classical pairwise approach.

---

> > ### Author Response · Authors · 2022-11-13
> > **Response to Reviewer NRwE (Part 2 / 3)**
> >
> > >**Q3:**  if this N-trajectories idea can apply to other scenarios since most previous works use pairwise trajectories.
> >
> > **A:**  Thanks for the great suggestions. Indeed we believe our idea of using N-trajectories simultaneously to train the reward function can be applied to other scenarios where pairwise-preference learning is used. These applications include but not limited to
> > - Imitation Learning [1,2],
> > - Reward function learning with human feedback (applications such as instructed pretraining / fintuning) [3].
> >
> > Our intuition is that as long as the preference scores/orderings for multiple trajectories are available, using our N-trajectories idea can be more efficient for learning effective reward functions.
> >
> > One limitation of our approach, however, is that when applying our methods to the human preferences (e.g, applications like the Instruct GPT), sometimes human workers may not give accurate labels/annotations when facing many choices, i.e., human workers may find it difficult to rank several alternatives.
> > Nevertheless, we believe the issue is more related to the literature of crowdsourcing, which is beyond the scope of this paper.
> >
> > [1] Brown, Daniel, et al. "Extrapolating beyond suboptimal demonstrations via inverse reinforcement learning from observations." International conference on machine learning. PMLR, 2019.
> >
> > [2] Brown, Daniel S., Wonjoon Goo, and Scott Niekum. "Better-than-demonstrator imitation learning via automatically-ranked demonstrations." Conference on robot learning. PMLR, 2020.
> >
> > [3] Ouyang, Long, et al. "Training language models to follow instructions with human feedback." arXiv preprint arXiv:2203.02155 (2022).
> >
> >
> >
> > >**Q4:**  Regarding the reproduction of CASPI, it is not clear which version from the official CASPI codebase did the author run to get the results in Table 1.
> >
> > **A:** We use the latest version of the CASPI, whose last commit is on Nov 16, 2021.
> > The original CASPI repository does not fix the random seed in learning the reward function, and thus we are unable to reproduce their results.
> > We fix this issue in our reproduction, and this is the only code modification we made.
> > We provided our CASPI results on each of the tested five random-seeds in Appendix D of the revised paper.
> >
> > >**Q5:**   It should be possible to show the Median score of the RewardNet and RewardMLE experiments and compare them to the CASPI original paper.
> >
> > **A:** Thanks for the suggestion!
> >
> > Due to the issue discussed in the previous question, we choose to compare our methods with our reproduction of the CASPI results.
> >
> > In Appendix D Table 5 of the revised paper, we provide a detailed breakdown of the scores onto each of the five tested random seeds, comparing CASPI and our two methods: RewardNet+GS, $N=3, \Phi=(\cdot)^1$ and  RewardMLE+GS, $N=5, \Phi=\exp(\cdot)$.
> >
> > We see that both our methods have higher Combined Scores than CASPI *on each of the five tested random-seeds*. This shows a stable improvement of our methods over CASPI, which aligns with our intuition discussed in Section 3 of our paper.
> >
> > In Appendix D of the revised paper, we additionally discuss why we choose to report the Average over the tested random-seeds, rather than the Median, in our experimental section.
> >
> >
> >
> > >**Q6:**   It is not mentioned how the authors choose the length of the trajectories to update the reward model.
> >
> > **A:**
> > Sorry for the confusion.
> > We do not change the length of the trajectories in the dataset.
> > The reward model is updated by the preference scores/orderings among multiple trajectories of the same length.
> > Our intuition is that trajectories of the same length may roughly correspond to tasks of similar complexity, making the preference among them more comparable and meaningful.
> > This approach is also taken by the prior work CASPI.

---

> > > ### Author Response · Authors · 2022-11-13
> > > **Response to Reviewer NRwE (Part 3 / 3)**
> > >
> > > >**Q7:**  If all of the dialogues have an equal length of trajectory, could it happen that the agent arrives at the goal earlier than the set length or need more steps to achieve it?
> > >
> > > **A:** Thanks for this interesting question!
> > > Please check the above response for our clarification on the trajectory length.
> > > Our assumption is that the preference scores/orderings should themselves reflect whether arriving at the goal earlier should get a bonus, and/or whether needing more steps to achieve the goal (incomplete task) should be punished. Our work seeks to learn a reward function that aligns with the preference among the multiple trajectories, which is orthogonal to the design and obtain of the preference scores/orderings.
> > >
> > > In our experiments of using the Combined Score to generate the trajectory ordering, an incomplete dialogue trajectory (i.e., the agent potentially needs more steps to achieve the goal) can be punished. This is because an incomplete task can lead to a lower Inform rate and/or a lower Success rate, both of which measure the dialogue-task completion and are components in calculating the Combined Score (see Section 5 of our paper). This punishment on incomplete tasks intuitively aligns with the usage of the TOD system.
> > >
> > > Using the Combined Score to generate the trajectory orderings may not factor in arriving at the goal earlier. Designing an automatic evaluation metric that explicitly takes this into account is out of the scope of this paper. We thank the reviewer again for raising this interesting question and will investigate this in our future work.
> > >
> > >
> > >
> > > >**Q8:**  Suggestions on typos, writings and the illustrating figure.
> > >
> > > **A:** Thank you very much for the kind suggestion! We have updated the paper and the Figure 1 according to your comments.

---

> ### Author Response · Authors · 2022-11-17
> **Following up with Reviewer NRwE**
>
> Dear Reviewer NRwE,
>
> Thank you for your careful review!
>
> We would like to double check if our response can address your concerns on our approach of using multiple dialogue-trajectories, the novelty and reproducibility of our work, and other related concerns in your review?
>
> Please kindly let us know if you have any remaining questions or any further concerns so that we can address them and further revise our manuscript during the discussion period.
> If our response and revised manuscript have addressed your concerns, please could you re-evaluate our work based on the updated information?

---

### Official Review · Reviewer_kmgW · 2022-10-24

**Confidence:** 3
**Correctness:** 3
**Technical Novelty And Significance:** 3
**Empirical Novelty And Significance:** 3
**Recommendation:** 8

**Clarity, Quality, Novelty And Reproducibility:**

Clarity: The paper is organized well, especially the ablation section (5.2).

Quality: The experiments and analysis seem of high quality.

Novelty: The approach builds upon prior work on learning TOD agents via rewards in RL, especially CASPI. The authors generalize the idea and demonstrate the effectiveness of the more general formulation(s)

Reproducibility: Could not identify whether code is to be released or if experiments are reproducible.

**Strength And Weaknesses:**

Strengths:
- RewardNet and RewardMLE are both demonstrated to out-perform SOTA algorithms
- The authors demonstrate that existing work (CASPI) can be framed as a specific case of their proposed reward algorithms (RewardNet to be precise), showing that their contributions are more general
- The ablation questions (5.2) are well-posed and answered in a logical fashion
- Human evaluation

Weaknesses:
- The number of trajectories seems to have a nontrivial effect on total score; in a real-world setting how well do e.g. the choices of N=3 for RewardNet and N=5 for RewardMLE perform? It would be helpful to see a discussion of parameter sensitivity.
- Would have liked to see more discussion in terms of a breakdown between multi-domain and dialogs from each separate domain in MultiWOZ, to investigate if there is a disparate effect of learning a TOD agent via RewardNet/RewardMLE depending on the complexity of the conversation (e.g. length, number of slots/actions) and the domain complexity (e.g. possible slot values, uniformity of distribution of slot values).

**Summary Of The Paper:**

This paper aims to design reward functions for training TOD agents through reinforcement learning, inspired by Learning-to-Rank (LTR) methods. The authors demonstrate that their RewardNet and RewardMLE methods achieve strong improved performance on MultiWOZ 2.0 compared to SOTA methods for training TOD agents.

**Summary Of The Review:**

This paper justifies its choice of approach to training TOD agents (via generalized rweard function designs) and effectively demonstrates their effectiveness in a multi-domain setting. I would have liked to see a more thorough discussion but the paper itself reads well.

---

> ### Author Response · Authors · 2022-11-13
> **Response to Reviewer kmgW**
>
> Thanks to the Reviewer kmgW for your time and valuable suggestions.
> We would like to bring to your attention our general response on the reproducibility of our work.
> The followings are our detailed responses to your comments:
>
>
> >**Q1:** The number of trajectories seems to have a nontrivial effect on the total score; how well do the choices of $N$ perform in real-world settings.
>
> **A:**
> We think that if high-quality preference orderings/scores among multiple dialogue trajectories are readily available, using a larger value of $N$ can be beneficial.
> In practice, however, such good preference orderings/scores may need to be collected.
> In fact, when applying our methods to the human preferences (e.g, applications like the Instruct GPT), the human workers may not give accurate labels/annotations when facing many choices, i.e., human workers may find it difficult to rank several alternatives.
> In this case, using a larger value of $N$ can degrade the data quality and can thus be harmful.
> Overall, for practitioners, we recommend to use a reasonably large value of $N$, such as 3 - 5.
>
> Besides, as our current and additional experiments shows, for different dataset variants (MultiWoz 2.0 and 2.1), and different backbone algorithms (MinTL and GALAXY), our methods with a reasonably large $N$ can outperform baseline algorithms that do not enjoy the benefits of our techniques. So in general we believe it is ok to choose $N \in [3,5]$ for new problems in real-world settings, which could generally provide benefits for a method that does not utilize rewards.
>
> As for the ablation study in our paper, it helps understand the ability and limitations of our proposed methods. We believe the conclusion of the ablation study does not conflict with the superior performance of our method with a reasonable choice of $N$.
>
>
>
>
> >**Q2:** More discussion in terms of a breakdown between multi-domain and dialogs from each separate domain in MultiWOZ, to investigate if there is a disparate effect of learning a TOD agent via RewardNet/RewardMLE depending on the complexity of the conversation
>
> **A:**  Thanks for the great suggestion. Due to the limit of page and time, currently our main focus is to validate the effectiveness of our proposed method on the overall performance of the TOD tasks. Therefore, our ablation study mainly focuses on studying the algorithmic variants, so that readers can get a more general sense of how good our method could be. We definitely think the suggestion is valuable, and we will investigate it as a future work!

---

> ### Author Response · Authors · 2022-11-17
> **Following up with Reviewer kmgW**
>
> Dear Reviewer kmgW,
>
> Thank you for your appreciation of our work!
>
> May we kindly ask if our response has addressed your concerns on the parameters in our method and on the reproducibility of our work?
> We will be more than happy to address any of your remaining/further questions during the discussion period, and possibly further revise our manuscript.

---

### Official Review · Reviewer_tRUe · 2022-10-25

**Confidence:** 4
**Correctness:** 3
**Technical Novelty And Significance:** 2
**Empirical Novelty And Significance:** 3
**Recommendation:** 6

**Clarity, Quality, Novelty And Reproducibility:**

Clarity: Most things are explained clearly either in the main paper or the appendix.
Quality: well-organized paper.
Novelty: The novelty is limited, details please refer to Weaknesses
Reproducibility: RL training is unstable. More experiment hyper-param setup is needed. The authors are better to provide source code to ensure reproducibility.

**Strength And Weaknesses:**

Strengths:
1. The paper is well-organized and easy to understand.
2. How to efficiently learn and utilize a reward function for training end-to-end task-oriented dialogue agents is fundamental and deserving problem.

Weaknesses:
1. The paper only conducts experiments on one benchmark MultiWOZ 2.0. As a methodological contribution, it will be more convincing to verify the effectiveness on more benchmarks.
2. The novelty is limited. How to efficiently learn the reward function in RL-based dialogue agents is a classic problem. Various existing literature[1][2][3] has discussed this problem and proposed the corresponding solution.
3. More implementation details are needed in the main paper, and I did not see the code.

References:
1. Curriculum Learning Based on Reward Sparseness for Deep Reinforcement Learning of Task Completion Dialogue Management
2. Playing 20 question game with policy-based reinforcement learning
3. Guided Dialog Policy Learning without Adversarial Learning in the Loop

**Summary Of The Paper:**

This paper investigates the question of how to efficiently learn and utilize a reward function for training end-to-end task-oriented dialogue agents. To be specific, the authors introduce two generalized objectives (RewardNet and RewardMLE) for reward-function learning, motivated by the classical learning-to-rank literature. Further, they also propose a stable policy-gradient method to guide the training of the end-to-end TOD agents. The proposed method achieves competitive results on the end-to-end response-generation task on the widely-used dialogue benchmark MultiWOZ 2.0.

**Summary Of The Review:**

I like the problem of efficiently learning the reward function in end-to-end TOD dialogue systems. My main concern is the limited technical novelty of the proposed RL method for a machine learning conference like ICLR. Moreover, considering the unclear reproducibility, I tend to give a weak rejection.

---

> ### Author Response · Authors · 2022-11-13
> **Response to Reviewer tRUe (Part 1 / 2)**
>
> We would like to thank Reviewer tRUe for the detailed review.
> Please check our general response on the novelty and the reproducibility of our work.
> The followings are our response to your questions:
>
>  > **Q1:** It would be more convincing to verify the effectiveness on more benchmarks.
>
> **A:**
> Thank you for the suggestions.
> In Appendix F of the revised version, we provide the results on the MultiWOZ 2.1 dataset, which is also a popular TOD dataset. On this dataset, our method again performs competitively with the baselines.
>
> To further demonstrate the effectiveness and general applicability of our proposed methods, we apply our reward learning and utilization methods to the recently proposed GALAXY [4] method, which is an end-to-end TOD-system learning method with competitive performance on the MultiWOZ 2.0 dataset. Appendix E of the revised paper provides a detailed breakdown of the scores onto each of the tested five random-seeds. In particular, adding our RewardMLE+GS method improves the average Combined Score of the vanilla GALAXY by 3.27, and adding our RewardNet+GS improves the vanilla GALAXY by 2.11. We believe that these additional benchmarking results can further corroborate our methodological contribution.
>
>
> [4] He, Wanwei, et al. "Galaxy: A generative pre-trained model for task-oriented dialog with semi-supervised learning and explicit policy injection." Proceedings of the AAAI Conference on Artificial Intelligence. Vol. 36. No. 10. 2022.
>
> > **Q2:**  How to efficiently learn the reward function in RL-based dialogue agents is a classic problem. Existing literature has discussed the problem and proposed the corresponding solutions.
>
> **A:**
> We believe our proposed reward learning and utilization method is fundamentally different from the prior works [1,2,3], which only deal with the dialogue management sub-task that is much simpler than our setting of directly generating  human-language-like system responses.
> The decomposition of the entire TOD problems into several sub-tasks, which include the dialogue management, requires intensive structural designs, as discussed in the second paragraph of Section 1 in our paper.
>
> By contrast, our approach can be directly applied to many existing end-to-end TOD training methods (e.g., MinTL and GALAXY), where the agents can leverage some pre-trained language models to directly generate human-language-like system responses, without manually decomposing the TOD task as in the previous approaches.
> Due to the difficulty of response generation compared to dialogue management, we believe the prior works [1,2,3] listed here are not directly applicable to our setting of end-to-end TOD-training. To further clarify the differences, we add a detailed discussion of the prior works [1,2,3] in Appendix C of the revised version.
> For some more closely-related prior works, such as CASPI and GPT-Critic, we have discussed them in our related-work section and compared against them in our experiments.
>
> Based on the above reasons, on the Q1 in our general responses, and on the additional discussion in Appendix C, we kindly disagree with the concern on our limited novelty.
>
>
> [1] Saito, Atsushi. "Curriculum learning based on reward sparseness for deep reinforcement learning of task completion dialogue management." Proceedings of the 2018 EMNLP workshop SCAI: The 2nd international workshop on search-oriented conversational AI. 2018.
>
> [2] Hu, Huang, et al. "Playing 20 question game with policy-based reinforcement learning." arXiv preprint arXiv:1808.07645 (2018).
>
> [3] Li, Ziming, et al. "Guided dialog policy learning without adversarial learning in the loop." arXiv preprint arXiv:2004.03267 (2020).
>
> > **Q3:**  More implementation details are needed in the main paper.
>
> **A:**
> Thanks for the great suggestion! We have added a section on implementation details in Appendix G of the revised paper.
>
> > **Q4:**  RL training is unstable.
>
>
> **A:**
> To reduce the instability and randomness, in the experimental section, we report the average scores across 5 different random-seeds whenever applicable.
>
> In Appendix D of the revised paper, we provide a detailed breakdown of the scores onto each of
> the five tested random seeds, comparing CASPI and our two methods: RewardNet+GS, $N=3, \Phi=(\cdot)^1$ and  RewardMLE+GS, $N=5, \Phi=\exp(\cdot)$.
> We see that the performances of our methods are relatively stable across random seeds.  Further, both our methods have higher Combined Score than CASPI *on each of the five tested
> random-seeds*. This shows a stable improvement of our methods over CASPI, which aligns with our intuition discussed in Section 3 of the paper.
>
> Therefore, we kindly argue that our method does not suffer much from the unstable training.

---

> > ### Author Response · Authors · 2022-11-13
> > **Response to Reviewer tRUe (Part 2 / 2)**
> >
> > >**Q5:**  More experiment hyper-param setup is needed.
> >
> > **A:**
> > In Section 5.2, we have conducted detailed ablation studies on important hyperparameters: the number of trajectories used for the reward-learning losses, the probabilistic transforms in the reward learning objective, the weight for the policy-gradient optimization of the response-generation model. We analyzed how these hyperparameters would affect the final performance. Further, we conducted a toy experiment showing the advantage of our choice of the Gumbel-softmax trick over the classical REINFORCE method for policy-gradient update.
> >
> >
> > We left most of the common hyperparameters unchanged from the prior work, such as the learning rate and batch size. Please kindly let us know what other hyperparameters you think are important to demonstrate the effectiveness of our approach.

---

> ### Author Response · Authors · 2022-11-17
> **Following up with Reviewer tRUe**
>
> Dear Reviewer tRUe,
>
> Thank you for your constructive review!
>
> We hope that our response can address your concerns.
> We would like to kindly confirm if you still have concerns on the applicability of our method, on the novelty and reproducibility of our work, or have any further concerns.
> If so, we would like to take the discussion period to respond to any of your remaining/new concerns and address them in our revision.
> If our response and revised manuscript have answered your questions, would you mind re-evaluating our work based on the updated information?

---

> ### Comment · Reviewer_tRUe · 2022-11-29
> **Thanks**
>
> Thanks for your detailed responses. The responses address most of my concerns and I raise my score.

---

### Official Review · Reviewer_7ZSu · 2022-11-12

**Confidence:** 3
**Correctness:** 3
**Technical Novelty And Significance:** 3
**Empirical Novelty And Significance:** 2
**Recommendation:** 6

**Clarity, Quality, Novelty And Reproducibility:**

The method is quite complicated and makes use of many parameterized tricks.  Although the novel contribution is a small deviation from existing work, getting it all working seems very involved and I think there will be significant reproducibility concerns if the code is not shared.  Are there plans to release the code?  I would have to consider lowering my score if the code was not going to be released or at least distributed by the authors upon request.

Some questions regarding clarity and soundness:
What is going on with BLEU as it relates to Inform/Success?  Especially in a case like the Fig 2 10%.

Maybe I missed this, but the description addresses the case where trajectory lengths are equal.  How does this work in practice?

I thought it was less than ideal that PAR and AEST pop up in Fig 1, and are basically not integrated with the written discussion.

Typos / Grammar:

(b): Does different -> Do difference

protocal -> protocol

**Strength And Weaknesses:**

Strengths:
The empirical results of this paper make a compelling case for its acceptance -- the method achieves a new state-of-the-art on the MultiWOZ 2.0 dataset, improving over the closely related CASPI method by ~1-1.5 pts in success and inform metrics.  In low data settings, the margins are quite large.  The analyses are generally good, and show advantages of this method in many settings.

Weaknesses:
Working against the paper is perhaps the similarity between previously best method, and concerns about the strengths of the results.  The relatively small margins of improvements in main evaluation are gained only in certain hyperparameter settings.  If one were to choose an architecture with a given N or \Phi in advance, there appears to still be a relatively good chance that there will be very small improvements (< 0.5, or even < 0.1) on some measures, to an extent that it would probably be insignificant.  Rather than tune these on some sort of dev set to mimic a realistic scenario, different runs get different rows in the table, and are presented in a manner of, ~"Look, one of our methods will always win by a good margin", only that which system that is is unfortunately not consistent.  I find this pretty disappointing.  It is saved primarily by the fact that usually all of these systems outperform (by microns) the previous SOTA, but it is not beyond reason to wonder if the case would be true for even more values of N, or on a different dataset or task.  This is an obvious place where the paper could be improved.

This relates to another weakness -- main results are averaged, and authors don't report the variance of their method.  The authors state "overall performance significantly improves over CASPI", but we are not given access to statistical significance tests / confidence margins over multiple runs.  These would be good to have in general, but the former is necessary to make claims of significance, and if that is not the authors' intent then it should be reworded.

The (automatic evaluation) results on MultiWOZ are promising, but the above issues make it difficult to have great faith in the results, given they are applied just to this dataset.  It would have been better to apply the method on a couple of other popular TOD datasets.  It seems KVRET and SGD benchmark are similar in spirit to MultiWoz.  The results presented here are maybe just about/above the threshold of publication, but there's certainly value in asking for additional experiments and metrics.


**Summary Of The Paper:**

 (To the authors, apologies for the delays - this was an emergency review taken on right before an unexpectedly busy week(s))

 In this work the authors explore ways in which to learn a reward function for task-oriented dialogue systems, proposing two new objectives phrased within the context of a learning-to-rank objective.  Given a set of scored (and hence ranked) TOD trajectories, the aim is to learn local reward functions whose accumulated scores reflect the original ranking.  They propose two methods, one doing just that (RewardMLE), the other minimizing the cross entropy between the two scores (RewardNet).  The learned reward function is incorporated into the training objective using Gumbel-softmax to reduce variance from the vanilla REINFORCE estimator.  On one benchmark dataset, variations of the proposed approach improve on standard TOD evaluation metrics, and show strong improvements in artificial data sparsity and human evaluations.

**Summary Of The Review:**

I think the existing experiments are probably sufficient for publication if I had to come down one way or another, but they do leave a lot to be desired.  The method seems to be a useful generalization of existing reward learning for TOD and one that could be applied in many situations.  If the authors can show that the performance gains are consistent, and that high performance model configurations are somewhat identifiable in advance, then there is value in the method.

---

> ### Author Response · Authors · 2022-11-13
> **Response to Reviewer 7ZSu (Part 1 / 3)**
>
> Many thanks to Reviewer 7ZSu for the comprehensive review!
> We would like to first bring to your attention our general response on the novelty and the reproducibility of our work.
> The following are our responses to your questions.
>
>  > **Q1:** Main evaluation only shows relatively small margins of improvements in certain hyperparameter settings.
>
> **A:**
> We respectfully disagree that our main evaluation (Table 1) has only “small margins of improvements.”
> As discussed in Section 1 of our paper, the main focus of our work is to discuss methods to properly learn and utilize an intermediate reward function for training end-to-end TOD systems.
> Table 1 shows that compared with prior works that do not utilize an intermediate reward function, our methods with Combined Score $> 106$ are clearly better than the best such baseline scores of $101.13$ (GPT-Critic).
> This shows the importance of learning and utilizing an intermediate reward function.
>
> Compared with the prior work CASPI that can be viewed as a special instance of our framework, our methods show an improvement of $\approx 2.0$ in the overall Combined Score, with improvements on each of the detailed evaluation metrics: Inform rate, Success rate and the BLEU score. In particular, compared with CASPI, our methods can improve the task completion by $> 1.26$ in Inform rate and $> 1.1$ in Success rate, without sacrificing the fluency of the generated responses (the BLEU score).
> Further, as discussed in our ablation study Section 5.2 (c), with additional fine-tuning, our methods can achieve an even higher Combined Score of $\approx 107.2$, which is $> 2.4$ higher than CASPI.
> Apart from the automatic evaluations in Table 1, in Section 5.3 Figure 6, we showed the results of human evaluation, comparing our model, CASPI, and GPT-Critic. It is clear that our method outperforms these two baselines in both the appropriateness and fluency of the generated response.
> Finally, in Section 5.3, we conducted two case studies comparing the generated responses from our method with the baselines CASPI and GPT-Critic.
> It is clear from Appendix A Table 3 and 4 that these two baselines can fail to continue the conversation, while the generated responses from our model show higher quality.
> To sum up, based on the automatic evaluations in Table 1, the human evaluation and case studies in Section 5.3, we believe that our methods have sufficient improvements over the prior works.
>
> Please check below our response to Q3 for our presentation flow of the main results, and our response to Q5 for a detailed breakdown comparison between our methods and CASPI on each of the tested five random seeds.
>
> In Section 5.2, we conducted ablation studies on several key hyper-parameters in our methods, from which we see that our methods are generally robust to the choice of those hyper-parameters. Specifically, in Fig. 2a and 2b, all variants have Combined Scores above 103.5, which is highly competitive to the baselines in Table 1. This shows that our method is generally robust to the number of trajectories used.
> In Section 5.2 (b), we see that our RewardMLE model is relatively insensitive to the choice of probabilistic transform in the reward learning objective -- all the considered variants have a competitive Combined Score of at least 104.
> Further, in Section 5.2 (c), we show that our model can be relatively robust to the choice of $\alpha$, the weight for the policy-gradient optimization of the response-generation model.
> Based on these ablation studies, we believe that our high-performing model configurations can be somewhat identifiable in advance.
>
> > **Q2:**  If were to choose an architecture with a given N or $\Phi$, one would probably get very small improvements on some measures.
>
> **A:**
> Apart from our response to Q1, we would like to gently note that the Combined Scores of our methods in Table 1 show that our methods *overall* perform better than previous state-of-the-art methods. In fact, even though the improvements on some metrics can be relatively small, the gain on other metrics can be relatively big, making the overall performance-measure (Combined Score) of our method better.
>
> We also showed in the human evaluation and case studies (Section 5.3) that our methods can improve the subjective perception of response quality over the baselines.

---

> > ### Author Response · Authors · 2022-11-13
> > **Response to Reviewer 7ZSu (Part 2 / 3)**
> >
> > > **Q3:**  Different runs get different rows in Table 1 and the best method on each metric is not consistent.
> >
> > **A:**
> > Sorry for the confusion. We would like to clarify that we do not aim at finding a variant that performs the best among all the detailed evaluation metrics: Inform rate, Success rate and the BLEU score. As shown in Table 1, this is not achieved in some baseline methods. For example, SOLOIST has a BLEU score of 16.54, which is lower than the BLEU score 16.90 of a much earlier baseline SFN + RL.
> >
> > Further, even though some of our variants are not the best in certain evaluation metrics, their scores are still better than the best baselines. For example, our  RewardNet+GS, $N=3, \Phi=(\cdot)^1$ has an Inform rate of 92.63 and a BLEU score of 18.35, both of which are not the best scores but both are clearly better than the best baselines.
> >
> > We present each of the four variants in a different row in Table 1 to show that:
> > 1. The benefit of using more than two trajectories to learn the reward model (the first two variants), corresponding to our two generalized objectives for reward learning in Section 3.1.
> > 2. The efficacy of directly optimizing the response-generation model *w.r.t.* the learned reward function via the Gumbel-softmax trick (the last two variants, indicated by "+ GS"), which corresponds to stable policy-gradient update of the response-generation model in Section 3.2.
> >
> > Based on these, we give different variants a different row in Table 1 for a more complete discussion of our proposed methods.
> >
> > > **Q4:**  Will the system perform well for even more values of N, or on a different dataset or task.
> >
> > **A:**
> > As shown in Section 5.2 (a), our method is generally robust to the number of trajectories. In fact, as shown in Figs. 2a and 2b, all variants have Combined Scores above 103.5, which is highly competitive to the baselines in Table 1.
> > In particular, the RewardNet variant with $N = 9$ trajectories has a Combined Score of 105.8, and the RewardMLE variant with $N = 11$ trajectories has a Combined Score of 106.3.
> > These two results are both better than the best baseline score of 104.78 (CASPI) in Table 1. This shows that even larger values of N can still be helpful for the performance.
> >
> > In Appendix F of the revised version, we provide the results on the MultiWOZ 2.1 dataset, which is also a popular TOD dataset. On this dataset, our method again performs competitively with the baselines.
> >
> > To further demonstrate the effectiveness and general applicability of our proposed methods, we apply our reward learning and utilization methods to the recently proposed GALAXY [1] method, rather than the MinTL backbone used for our main experimental results (Section 5).
> > The GALAXY method is also an end-to-end TOD-system learning method, and achieves competitive performance on the MultiWOZ 2.0 dataset. Appendix E of the revised paper provides a detailed breakdown of the scores onto each of the tested five random-seeds. In particular, adding our RewardMLE+GS method improves the average Combined Score of the vanilla GALAXY by 3.27, and adding our RewardNet+GS improves the vanilla GALAXY by 2.11. We believe that these additional benchmarking results can further corroborate the efficacy of our methods.
> >
> > [1] He, Wanwei, et al. "Galaxy: A generative pre-trained model for task-oriented dialog with semi-supervised learning and explicit policy injection." Proceedings of the AAAI Conference on Artificial Intelligence. Vol. 36. No. 10. 2022.
> >
> > > **Q5:** Only reported averaged main results without the variance. The sentence "overall performance significantly improves over CASPI" is inappropriate.
> >
> > **A:**
> > Thanks for the suggestion!
> >
> > We kindly note that we follow the baselines [e.g., 2, 3, 4] to report the average scores without the variance.
> >
> > In Appendix D Table 5 of the revised paper, we provide a detailed breakdown of the scores onto each of the five tested random seeds, comparing CASPI and our two methods: RewardNet+GS, $N=3, \Phi=(\cdot)^1$ and  RewardMLE+GS, $N=5, \Phi=\exp(\cdot)$.
> >
> > We see that both our methods have higher Combined Scores than CASPI *on each of the five tested random-seeds*. This shows a stable improvement of our methods over CASPI, which aligns with our intuition in Section 3 of our paper and the main experimental results in Section 5.1 of the paper.
> >
> > Finally, thank you for pointing out the referred sentence. We have reworded this sentence per your suggestion.
> >
> > [2] Zhang, Yichi, Zhijian Ou, and Zhou Yu. "Task-oriented dialog systems that consider multiple appropriate responses under the same context." Proceedings of the AAAI Conference on Artificial Intelligence. Vol. 34. No. 05. 2020.
> >
> > [3] Lin, Zhaojiang, et al. "Mintl: Minimalist transfer learning for task-oriented dialogue systems." arXiv preprint arXiv:2009.12005 (2020).
> >
> > [4] Jang, Youngsoo, Jongmin Lee, and Kee-Eung Kim. "GPT-Critic: Offline Reinforcement Learning for End-to-End Task-Oriented Dialogue Systems." ICLR 2021.

---

> > > ### Author Response · Authors · 2022-11-13
> > > **Response to Reviewer 7ZSu (Part 3 / 3)**
> > >
> > > > **Q6:**  Tested dataset is limited. Better to try on other datasets such as KVRET and SGD.
> > >
> > >
> > > **A:**
> > > Thank you for this great suggestion.
> > > Please see our previous response to Q4 for discussions on our additional experiments on (1) the MultiWOZ 2.1 datasets; and (2) applying our methods onto another backbone GALAXY.
> > > We believe that these additional experiments can further validate our methods.
> > >
> > > We deeply appreciate your pointing out some other TOD datasets. We have to admit that it takes some time to look into the suggested datasets, to pre-process the dataset, and to obtain benchmarking results. Given the limited time of the rebuttal period, we unfortunately have to leave them for future work.
> > >
> > > > **Q7:**  What is going on with BLEU as it relates to Inform/Success? Especially in a case like the Fig 2 10%.
> > >
> > > **A:**
> > > Thanks for raising this interesting question!
> > > We would like to confirm if you refer to the BLEU score of our model in Table 2 10% case. As discussed in the second paragraph in Section 5 of the paper, the BLEU score measures the fluency of the generated response while the Inform/Success rate measures the dialogue-task completion. We are unaware of any explicit relationship between BLEU and Inform/Success rate.
> > >
> > > We agree that the BLEU of our models in the 10% case is lower than the MinTL baseline. However, our Inform and Success rates are both better than the baselines, by a relatively wide margin. In the low resource setting, such as 10% of the training data, it may not be possible to achieve both good dialogue-task completion and good fluency in the generated response.
> > > It is possible that under this setting, the optimization routine of our methods choose to optimize the dialogue-task completion, while the MinTL chooses to optimize the fluency.
> > > We believe that our method is reasonable since the task-oriented dialogue (TOD) system is used to interact with the users to complete tasks.
> > >
> > > We also note that the BLEU scores of our methods under 10% training data are better than the scores under 5% training data, which is reasonable and as expected.
> > >
> > > > **Q8:**  The description addresses the case where trajectory lengths are equal. How does this work in practice?
> > >
> > > **A:** We agree with the Reviewer that it is theoretically possible to learn the reward function from the preference among trajectories of unequal lengths.
> > > However, we choose to update the reward model by the preference scores/orderings among multiple trajectories of the same length.
> > > Our intuition is that trajectories of the same length may roughly correspond to tasks of similar complexity, making the preference among them more comparable and meaningful.
> > > The prior work CASPI also takes this approach.
> > > For implementation, one can bucket the dialogue trajectories in the training set by the length and sample from a bucket for each update of the reward function.
> > >
> > >
> > > > **Q9:**  Suggestions on Figure 1 and typos.
> > >
> > > **A:** Thank you so much for the invaluable suggestions. We have updated Figure 1 and fixed the typos in the revised version according to your suggestions.

---

> ### Author Response · Authors · 2022-11-17
> **Following up with Reviewer 7ZSu**
>
> Dear Reviewer 7ZSu,
>
> We deeply appreciate your thoughtful review and your time, and hope that our response can address your concerns.
>
> May we kindly ask if you still have concerns on the empirical performance of our method, the applicability of our approach, or have any further concerns?
> We will be more than happy to address your remaining or new concerns and possibly revise our manuscript during the discussion period.
> If our response and revised manuscript have addressed your concerns, would you mind considering re-evaluating our work based on the updated information?

---

### Author Response · Authors · 2022-11-13
**General Response**

We would like to thank all reviewers for the valuable suggestions. We uploaded a new version of the paper and highlighted the changes in red. The revisions are:
- Add a discussion on some other reward-learning methods in RL-based dialogue agents (Appendix C).
- Add the per random-seed scores of our method and the baseline CASPI (Appendix D).
- Apply our proposed methods onto another backbone method GALAXY (Appendix E)
- Test our methods on a new dataset MultiWOZ 2.1 (Appendix F).
- Provide additional implementation details (Appendix G).
- Update the flow chart (Figure 1) and fix some typos.
- We release our code and model checkpoints at [https://anonymous.4open.science/r/Reward_Learning_Dialog-AC78/](https://anonymous.4open.science/r/Reward_Learning_Dialog-AC78/).

Below are our responses to some common questions from the reviewers.

> **Q1:**  Novelty

**A:**
Our paper studies how to properly learn and utilize a reward function for training *end-to-end* task-oriented dialogue (TOD) agents, which directly outputs human-language-like system response.

For training the reward function, our paper is one of the few, if any other exists, that manages to leverage the preference among more than two trajectories for reward learning.
We connect reward-function learning for the TOD system with learning-to-rank methods and with reward-learning-from-preferences in reinforcement learning.
To the best of our knowledge, this connection is novel to the community.
In fact, our work generalizes prior methods in RL that learn rewards from the pairwise preference [1,2,3].
We provided intuition for using multiple trajectories in Section 3.1 of the paper, with empirical evidence in Section 5.
Additional discussions are available in our response to Review NRwE.

We appreciate Reviewer tRUe for pointing out some prior works on reward learning in RL-based dialogue agents [4,5,6]. These works, however, only deal with the TOD sub-task of high-level dialogue-management, which is much simpler than our setting of directly generating human-language-like responses. Therefore, we think that these methods may not be applicable to our setting. We discuss this in detail in our response to Reviewer tRUe and in Appendix C of the revised paper.

Apart from the reward-learning methods, our paper further considers more stable gradient-estimation based on the Gumbel-softmax trick for training the end-to-end TOD systems. To the best of our knowledge, there are few prior works that consider advanced gradient-estimation methods for stable training RL-based (end-to-end) TOD systems. We believe this topic is important for both academia and practitioners.

Based on the above discussions, we believe our work is novel to the field of TOD and is a valuable addition to the literature.


[1] Christiano, Paul F., et al. "Deep reinforcement learning from human preferences." Advances in neural information processing systems 30 (2017).

[2] Brown, Daniel, et al. "Extrapolating beyond suboptimal demonstrations via inverse reinforcement learning from observations." International conference on machine learning. PMLR, 2019.

[3] Brown, Daniel S., Wonjoon Goo, and Scott Niekum. "Better-than-demonstrator imitation learning via automatically-ranked demonstrations." Conference on robot learning. PMLR, 2020.

[4] Saito, Atsushi. "Curriculum learning based on reward sparseness for deep reinforcement learning of task completion dialogue management." Proceedings of the 2018 EMNLP workshop SCAI: The 2nd international workshop on search-oriented conversational AI. 2018.

[5] Hu, Huang, et al. "Playing 20 question game with policy-based reinforcement learning." arXiv preprint arXiv:1808.07645 (2018).

[6] Li, Ziming, et al. "Guided dialog policy learning without adversarial learning in the loop." arXiv preprint arXiv:2004.03267 (2020).



 > **Q2:** Reproducibility

**A:**
To help reproduce our results, we release the source code and model checkpoints [here](https://anonymous.4open.science/r/Reward_Learning_Dialog-AC78/). We detail in the `readme` therein how to download the data and run the experiments.

---

### Decision · Program_Chairs · 2023-01-20

**Decision:**

Accept: poster

**Justification For Why Not Higher Score:**

With the majority of the support being sympathetic and supportive, but slightly lukewarm, I hesitate to recommend a higher rating than Accept (poster). That said, I think the author rebuttal addresses a number of the issues raised by the reviewers, which the reviewers have elected not to respond to further or revise their scores based on. If one paper in my stack were to be bumped up, I would suggest this one.

**Justification For Why Not Lower Score:**

Reviewers unanimously support publication.

**Metareview: Summary, Strengths And Weaknesses:**

The case for acceptance of this paper defining and evaluating new reward functions for RL-based task-oriented dialogue is fairly clear. Reviewer consensus is uniformly on the side of acceptance, ranging from sympathetic to excited. The authors responded the the majority of reviewer concerns, and I would ideally have liked to see some further engagement from reviewers during this period to determine if they were willing to raise their scores. The main concern expressed during review seems to be surrounding the novelty of the method, which is typically a highly subjective and noisy line of criticism. From reading the author rebuttals, I don't think there should be any outstanding doubts regarding the proposition that this work is sufficiently novel to warrant publication, and thus I am happy to write in support thereof.

**Note From Pc:**

if the above contains the word "oral" or "spotlight" please see: "oral" presentation means -> notable-top-5% and "spotlight" means -> notable-top-25%. As stated in our emails, we are disassociating presentation type from AC recommendations